# AN EFFICIENT SUBGRAPH GNN WITH PROVABLE SUBSTRUCTURE COUNTING POWER

## ABSTRACT

Enhancing the representation power of graph neural networks (GNNs) through their ability to count substructures is a recent trend in graph learning. Among these works, a popular way is to use subgraph GNNs, which decompose the input graph into a collection of subgraphs and enhance the representation of the graph by applying GNN to individual subgraphs. Although subgraph GNNs are able to count complicated substructures, they suffer from high computational and memory costs. In this paper, we address a non-trivial question: can we *count substructures efficiently and provably* with GNNs? To answer the question, we first theoretically show that the distance to the rooted nodes within subgraphs is key to boosting the counting power of subgraph GNNs. We then precompute structural embeddings that encode such information to avoid extracting information over all subgraphs via GNNs repeatedly. Experiments show that the proposed model can preserve the counting power of subgraph GNNs while running orders of magnitude faster.

## 1 INTRODUCTION

Message Passing Neural Networks (MPNNs) are the most commonly used Graph Neural Networks (GNNs). They have achieved remarkable success on graph representation learning (Kipf & Welling, 2017; Xu et al., 2018), and have been widely used in various downstream tasks (Wu et al., 2020; Zhou et al., 2020). However, the representation power of MPNNs is limited (Xu et al., 2018), thus increasing efforts have been spent on designing more powerful GNNs.

An essential and intuitive way to evaluate the representation power of GNNs is whether they **provably** approximate specific functions, such as counting graph substructures. Substructures often represent meaningful components in graphs and can reveal essential structural insights in chemistry (Deshpande et al., 2002; Jin et al., 2018), biology (Koyutürk et al., 2004), and sociology (Jiang et al., 2010). For instance, in a molecule graph, the presence of a 6-cycle (hexagon) passing through a node suggests its potential association with a benzene ring. Moreover, substructures are closely connected to many fundamental graph properties (Shervashidze et al., 2009; Preciado & Jadbabaie, 2010). In graph representation learning tasks, many targets are unknown or intractable functions of the graph structure which we need to learn from the data. However, MPNNs' substructure counting ability is shown to be very limited, failing to count even triangles (Chen et al., 2020). If the underlying graph functions depend on some substructures that the GNN theoretically cannot detect/count, we can never trust its performance on these tasks.

In this paper, we focus on the counting power of GNNs, i.e., whether GNNs can provably count the number of given connected substructures within a graph. Notice that we are not designing models to only count substructures (which can be perfectly done by traditional algorithms), but to propose GNNs with high enough representation power to solve unknown structure-related tasks that we need to learn from the data. In classic works (Fürer, 2017), to reach a high representation power, globally expressive models such as 3-WL (Maron et al., 2019; Tahmasebi et al., 2020) are needed. However, their high computational costs restrict their universal use in real-world applications.

Recently, a series of works called subgraph GNNs (You et al., 2021; Zhang & Li, 2021; Bevilacqua et al., 2022; Zhao et al., 2022; Frasca et al., 2022; Huang et al., 2023) are shown to provably count certain substructures. For an input graph, they first decompose the input graph into a collection of subgraphs (overlap is allowed) based on certain subgraph selection policies. Then some base GNNs are applied to the extracted subgraphs whose representations are used to enhance the graph

representation. Although faster than globally expressive GNNs, they still need to run GNNs over all subgraphs. Therefore they are much slower compared with classic MPNNs, and suffer from high computational cost when encoding large and dense graphs.

Based on the above observation, we raise a nontrivial question: can we count substructures **efficiently and provably** with GNNs, ideally using a similar cost to MPNNs? To answer the above, we decompose it into two sub-questions: (1) what provides the extra representation power of subgraph GNNs compared with classic MPNNs? (2) can we utilize such information efficiently without running GNNs on subgraphs? To answer the first question, we show that the key to boosting the counting power of subgraph GNNs is the **distance** to the rooted nodes within the subgraph. To answer the second question, we find that such distance information can be encoded into a precomputed structural embedding. Combining it with a base GNN, there is no need to run GNNs repeatedly on each subgraph. In this way, we only need to run GNNs on the original graph (augmented with precomputed structural embeddings), while being able to efficiently count substructures.

In summary, our contributions are listed as follows:

(1) We theoretically characterize the general substructure counting power of subgraph GNNs, showing that they are *much more efficient yet nearly as powerful* as globally expressive models.

(2) To accelerate subgraph GNNs, we theoretically show that the distance to the rooted nodes within subgraphs is key to boosting their representation power. We then propose a structural embedding to encode such distance information.

(3) We propose a model, Efficient Substructure Counting GNN (*ESC-GNN*), which enhances a basic GNN with the structural embedding. It only needs to run message passing on the whole graph, and thus is much more efficient than subgraph GNNs. We evaluate ESC-GNN on various real-world and synthetic benchmarks. Experiments show that ESC-GNN performs comparably with subgraph GNNs on real-world tasks and counting substructures, while running much faster.

## 2 RELATED WORKS

**Representation power of GNNs.** There are two major perspectives to evaluate the representation power of GNNs: the ability to distinguish non-isomorphic graphs, and the ability to approximate specific functions. In terms of differentiating graphs, existing works (Xu et al., 2018; Morris et al., 2019) showed that MPNNs are at most as powerful as 1-WL (Weisfeiler & Leman, 1968). Following works improve the expressiveness of GNNs by using high-order information (Morris et al., 2019; 2020; Maron et al., 2019; Bodnar et al., 2021b;a; Vignac et al., 2020) or augmenting node features (Bouritsas et al., 2022; Barceló et al., 2021; Dwivedi et al., 2021; Loukas, 2020; Abboud et al., 2021; Kreuzer et al., 2021; Lim et al., 2022).

In terms of approximating specific functions, some works use GNNs to approximate graph algorithms (Veličković et al., 2019; Xhonneux et al., 2021; Yan et al., 2022) or detect bi-connectivity (Zhang et al., 2023). In this paper, we focus on counting substructures. Previous works (Fürer, 2017; Arvind et al., 2020) relate the counting power to the expressiveness of GNNs. Following works count substructures with globally expressive networks (Murphy et al., 2019; Chen et al., 2020; Tahmasebi et al., 2020) that are with high computational costs. Several other works (Liu et al., 2020; Liu & Song, 2022; Yu et al., 2023) also focus on counting substructures. However, they do not provide theoretical guarantees for these counts. Therefore we cannot trust their performance on substructure-related tasks, which are abundant in chemistry and biology.

**Subgraph GNNs.** Subgraph GNNs can be divided according to their subgraph selection policies, such as graph element deletion (Bevilacqua et al., 2022; Cotta et al., 2021; Papp et al., 2021), k-hop subgraph extraction (Abu-El-Haija et al., 2019; Sandfelder et al., 2021; Nikolentzos et al., 2020; Feng et al., 2022), node identity augmentation (You et al., 2021), and rooted subgraph extraction (Zhang & Chen, 2018; Zhang & Li, 2021; Zhao et al., 2022; Frasca et al., 2022; Papp & Wattenhofer, 2022; Zhang et al., 2021; Huang et al., 2023; Qian et al., 2022). Most subgraph GNNs need to run message passing over all subgraphs, therefore performing much slower than classic MPNNs. This prevents their use in large real-world datasets.

**Positional/Structural Encodings on Graphs.** To leverage the spectral properties of graphs, many works (Dwivedi et al., 2020; Lim et al., 2022; Dwivedi et al., 2021; Kreuzer et al., 2021; Mialon et al.,

2021; Park et al., 2022; Rampášek et al., 2022) introduce the eigenvectors of the graph Laplacian as augmented node features. Other approaches introduce encodings with essential graph features (Li et al., 2020; Feldman et al., 2022; Ying et al., 2021; Wang et al., 2022; Yan et al., 2021; Bouritsas et al., 2022). Our work can be viewed as a subgraph-based structural encoding.

## 3 PRELIMINARIES

Let $G = (V, E)$ be a simple, undirected graph, where $V = \{1, 2, ..., N\}$ is the node set, and $E$ is the edge set. We use $x_v$ to represent the node attribute for $v \in V$, and $e_{uv}$ to represent the edge attribute for $uv \in E$. Denote the $h$-hop neighborhood of node $v$ as $V_v^h = \{u \in V | d(u, v) \leq h\}$, where $d(u, v)$ denotes the shortest path distance between node $u$ and $v$. For a special case where $h = 1$, we call it the neighborhood of node $v$: $N(v) = V_v^1$.

Define a subgraph of $G$ as any graph $G^S = (V^S, E^S)$ with $V^S \subseteq V$ and $E^s \subseteq E$. And an induced subgraph of $G$ is any graph $G^I = (V^I, E^I)$ where $V^I \subset V$, and $E^I = E \cap (V^I)^2$ is the set of all edges in $E$ where both nodes belong to $V^I$. For a $k$-tuple $\vec{v} = (v_1, ..., v_k) \in (V)^k$, define its rooted $h$-hop subgraph as $G_{\vec{v}}^h = (V_{\vec{v}}^h, E_{\vec{v}}^h)$, where $V_{\vec{v}}^h$ is the union of the $h$-hop neighborhoods of all vertices in $\vec{v}$: $V_{\vec{v}}^h = V_{v_1}^h \cup ... \cup V_{v_k}^h$, and $E_{\vec{v}}^h = E \cap (V_{\vec{v}}^h)^2$ are edges whose two nodes both belong to $V_{\vec{v}}^h$. Later, We will omit the hop parameter $h$ for simplicity.

In this paper, we focus on graph-level counting of connected substructures[1]. Connected substructures are substructures whose nodes belong to a connected component. We study four types of connected substructures that are widely used in existing works: cycles, cliques, stars, and paths. An $L$-path is a sequence of edges $[(v_1, v_2), ..., (v_L, v_{L+1})]$ such that all nodes are distinct; an $L$-cycle is an $L$-path except that $v_1 = v_{L+1}$; an $L$-clique is a fully connected graph with $L$ nodes; and an $L$-star denotes a set of edges $[(v, v_1), (v, v_2), ..., (v, v_{L-1})]$ where all nodes with different symbols are distinct. Two substructures are called equivalent if their sets of edges are equal. Given a substructure $S$ and a graph $G$, the **subgraph counting** is defined as counting the number of inequivalent substructures $C_S(S, G)$ that are subgraphs of $G$. The **induced subgraph counting** is defined as counting the number of inequivalent substructures that are induced subgraphs of $G$. In this paper, we focus on subgraph counting, but we also provide theoretical results on induced subgraph counting.

Following existing works (Chen et al., 2020; Huang et al., 2023), we formally define our task as:

**Definition 3.1.** Let $\mathcal{G}$ be the set of all graphs and $\mathcal{F}$ be a function class over graphs. We say $\mathcal{F}$ can count connected substructure $S$ on $\mathcal{G}$ if for all $G_1, G_2 \in \mathcal{G}$ such that $C_S(S, G_1) \neq C_S(S, G_2)$, their exists $f \in \mathcal{F}$ that $f(G_1) \neq f(G_2)$.

Using the Stone-Weierstrass theorem, the definition is equivalent to approximating subgraph-counting functions (Chen et al., 2020). If replacing $C_S$ by $C_I$, then the task will be naturally turned to induced subgraph counting.

## 4 COUNTING POWER OF SUBGRAPH GNNS

Subgraph GNNs have been proven to possess the capability to count specific substructures (Chen et al., 2020; You et al., 2021; Zhao et al., 2022; Huang et al., 2023). We commence by introducing subgraph GNNs in Section 4.1. In Section 4.2, we characterize the general substructure counting power of subgraph GNNs, by proving that they are nearly as powerful as globally expressive models while running much faster. We theoretically show that the distance information within the subgraph is key to boosting the counting power of GNNs. Such information can be encoded into a structural embedding, providing the basis for our proposed efficient substructure-counting model[2].

### 4.1 SUBGRAPH GNNS

In this section, we introduce subgraph GNNs and refer readers to the supplementary material for details on the Weisfeiler-Leman algorithm (WL) (Weisfeiler & Leman, 1968) and MPNNs.

---

[1] Some works (Huang et al., 2023) focus on node-level counting, which can also be transferred to graph-level counting.

[2] The distance-related information also strongly influences the expressiveness of subgraph GNNs since the subgraph selection policies and the unsymmetric treatment to the rooted node and other nodes are both based on such information.

Subgraph GNNs first represent the input graph by a collection of subgraphs based on certain subgraph selection policies. They then encode the subgraphs using backbone GNNs and aggregate subgraph representations into the graph representation. We note that there exist some other variants of subgraph GNNs (Qian et al., 2022; Zhao et al., 2022; Frasca et al., 2022; Bevilacqua et al., 2022), but in this paper, we focus on a specific type of subgraph GNNs without information exchange between subgraphs, which covers reconstruction GNNs (Papp et al., 2021; Cotta et al., 2021), ID-GNNs (You et al., 2021), and nested GNNs (Zhang & Li, 2021; Huang et al., 2023). We will show that this type of subgraph GNNs is powerful enough in terms of counting connected substructures.

We call their subgraph selection policy as *rooted subgraph extraction policy*. Existing works select subgraphs rooted at either nodes (Zhang & Li, 2021; You et al., 2021) or $k$-tuples (Huang et al., 2023; Qian et al., 2022), and typically use a 1-WL equivalent GNN as the backbone. In this paper, we propose a more general framework with *$m$-WL (or its equivalent GNN) as the backbone on subgraphs rooted at connected $k$-tuples*, i.e., $k$-tuples in which their induced subgraphs are connected. For a connected $k$-tuple $\vec{v}$, the selected subgraph is its rooted subgraph $G_{\vec{v}} = (V_{\vec{v}}, E_{\vec{v}})$.

Denote $c_{\vec{v},\vec{u}}^{t}$ as the color for an $m$-tuple $\vec{u} = (u_1, ..., u_m)$ in $G_{\vec{v}}$ at iteration $t$. It is computed by:

$$c_{\vec{v},\vec{u}}^{t} = \text{HASH}(c_{\vec{v},\vec{v}}^{t-1}, c_{\vec{v},\vec{u}}^{t-1}, c_{\vec{v},\vec{u},(1)}^{t}, ..., c_{\vec{v},\vec{u},(k)}^{t}) \tag{1}$$

where

$$c_{\vec{v},\vec{u},(i)}^{t} = \{\!\!\{ c_{\vec{v},\vec{q}}^{t-1} | \vec{q} \in N_{\vec{v},i}(\vec{u}) \}\!\!\}, i \in [m] \tag{2}$$

Here $N_{\vec{v},i}(\vec{u}) = \{(u_1, ..., u_{i-1}, w, u_{i+1}, ..., u_m) | w \in V_{\vec{v}}\}$ denotes the $i$-th neighborhood of $\vec{u}$ in $G_{\vec{v}}$.

Denote the final iteration as iteration $T$. The color of $\vec{v}$ after the final iteration will be the combination of all $m$-tuples' colors inside $G_{\vec{v}}$. Formally,

$$c_{\vec{v}} = Readout(\{\!\!\{ c_{\vec{v},\vec{u}}^{T} | \vec{u} \in (V_{\vec{v}})^m \}\!\!\}) \tag{3}$$

where $Readout$ is a readout function, e.g., the sum function. Intuitively, compared with MPNNs, subgraph GNNs (1) update the representation using not only the neighboring information but also the information from the rooted nodes; (2) the neighborhood is restricted to the subgraph level.

## 4.2 COUNTING POWER OF SUBGRAPH GNNS

Subgraph GNNs have long been used to count substructures. Existing works mainly focus on counting certain types of substructures, e.g., walks (You et al., 2021) and cycles (Huang et al., 2023) and do not relate subgraph GNNs with substructure counting in a holistic perspective. In this section, we characterize the general substructure counting power of subgraph GNNs by showing that they are nearly as powerful as globally expressive models, e.g., high-dimensional WL, while running much faster. We first theoretically characterize $m$-WL's power for counting **any** substructures.

**Counting power of $m$-WL.** Different substructures with no more than $m$ nodes have different initial isomorphic types. We can assign each isomorphic type a unique color, and define the color histogram of the graph as the output function. Therefore the lower bound of the counting power of $m$-WL is:

*Remark* 4.1. $m$-WL ($m \geq 2$) can count all connected substructures with no more than $m$ nodes.

As for the upper bound, we show that for any $m \geq 2$, there exists a type of connected substructure with $m + 1$ nodes that $m$-WL cannot count. The theorem is formally stated below, and the proofs of the following theorems are provided in the supplementary material.

**Theorem 4.2.** *For any $m \geq 2$, there exists a pair of graphs G and H, such that G contains an $(m + 1)$-clique as its subgraph while H does not, and that $m$-WL cannot distinguish G from H.*

Theorem 4.2 makes the lower bound in Remark 4.1 tight.

**Decomposition of counting connected substructures.** Before discussing the counting power of subgraph GNNs, we first provide the basis for the discussion: any graph-level substructure counting can be naturally decomposed into a collection of local substructure counting. For example, to count 3-cliques/3-cycles in a given graph, we can first compute the number of 3-cycles that pass each node, sum the number among all nodes, and then divide the number by 3 [3] to compute the graph-level result. We can extend the observation to a more general version:

---

[3]This number is strongly related to graph automorphism and is specific to the type of structure.

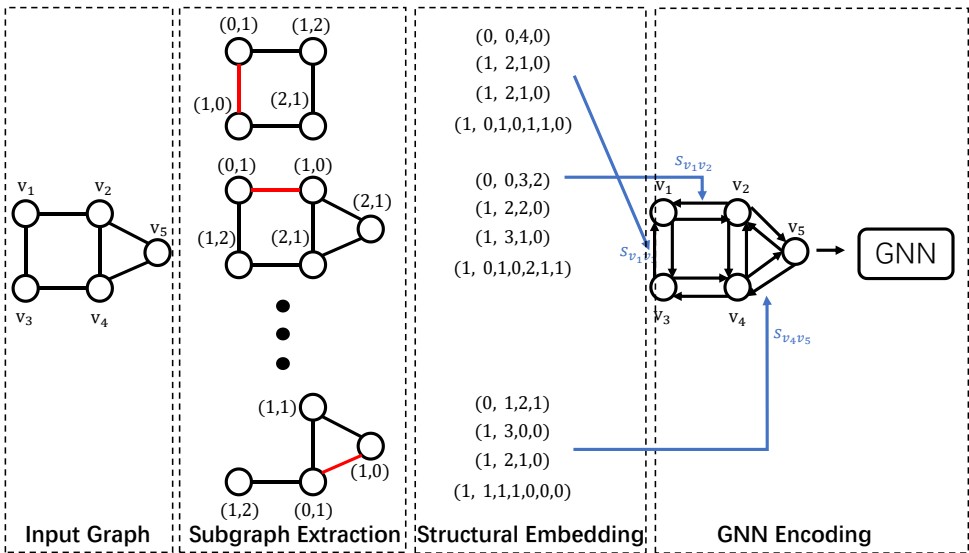

Figure 1: Framework of ESC-GNN. The rooted 2-tuples are colored red during subgraph extraction.

*Remark* 4.3. To count a certain type of connected substructures with no more than $m + k$ ($m \geq 2, k > 0$) nodes in a given graph, we can decompose it into counting over $k$-tuples. First, select a specific type of connected $k$-tuple whose induced subgraph is a subgraph of the target substructure. Then count the substructures that pass each $k$-tuple. Finally, the result is the sum of the numbers over all $k$-tuples divided by a constant dependent on the substructure.

**Counting power of subgraph GNNs.** We first give the lower bound below.

**Theorem 4.4.** *For any connected substructure with no more than $m + k$ ($m \geq 2, k > 0$) nodes, there exists a subgraph GNN rooted at $k$-tuples with backbone GNN as powerful as $m$-WL that can count it.*

Based on the proof and the insights gained from popular subgraph GNNs, we can safely conclude that *the distance from the rooted nodes to nodes within the subgraph provides valuable information for substructure counting*. As for the upper bound of the counting power of subgraph GNNs, existing works (Geerts, 2020; Frasca et al., 2022) show that for $m \geq 2$, (1) $m$-WL is as powerful as a specific GNN, called $m$-IGN (Maron et al., 2018); (2) a subgraph GNN rooted at $k$-tuples with backbone GNN as powerful as $m$-WL can be implemented by $(m + k)$-IGN. Therefore, we have:

**Proposition 4.5.** *Any subgraph GNN rooted at $k$-tuples with backbone GNN as powerful as $m$-WL ($m \geq 2$) is not more powerful than $(m + k)$-WL.*

Combining with Theorem 4.2, we obtain a tight characterization of subgraph GNNs' counting power for any substructures, which is the same as $(m+k)$-WL. This suggests that subgraph GNNs are nearly as powerful as globally expressive models in terms of graph-level general substructure counting. Note that globally expressive models might still be more powerful at counting specific substructures.

**Efficiency of subgraph GNNs.** The computational cost for $(m + k)$-WL is $O(|V|^{m+k})$, where $|V|$ denotes the number of nodes in the input graph. However, for subgraph GNNs rooted at $k$-tuples with backbone GNN as powerful as $m$-WL, the computational cost can be $O(|V|^k|V_s|^m)$, where $V_s$ is the largest number of nodes among all subgraphs. We point out that $|V_s|$ is usually much smaller than $|V|$. For example, when counting cliques and stars, the hop parameter can be set to 1; when counting cycles and paths, the hop parameter can be set to $m/2$. Therefore subgraph GNNs are much more efficient. In conclusion, subgraph GNNs rooted at $k$-tuples with backbone GNN as powerful as $m$-WL can reach a similar counting power to $(m + k)$-WL while being much more efficient. This can be a key motivation for the use of subgraph GNNs in counting substructures.

# 5 EFFICIENT SUBSTRUCTURE COUNTING GNN (ESC-GNN)

Despite the strong substructure counting ability, subgraph GNNs are still much slower than MPNNs since they need to run backbone GNNs on all subgraphs. In this section, we propose a model, Efficient Substructure Counting GNN (ESC-GNN), which can count substructures efficiently and effectively, while, more importantly, does not need to run GNN on subgraphs. ESC-GNN encodes the distance information within subgraphs into structural embeddings of edges in a preprocessing step. After that, it only needs to **run the backbone GNN on the input graph once rather than over all subgraphs**. We first introduce ESC-GNN, and then show its representation power theoretically.

## 5.1 FRAMEWORK OF ESC-GNN

**Basic framework.** In this section, we first introduce the framework of ESC-GNN, which is illustrated in Figure 1. In ESC-GNN, we adopt subgraphs rooted at 2-tuples, and use MPNN as the backbone GNN. Subgraph GNNs rooted at 2-tuples are more expressive than those rooted at nodes, but at the cost of even higher computational cost (Huang et al., 2023). Thus, instead of running backbone GNN over subgraphs to extract subgraph features (like distances), we directly encode them into some carefully designed structural embeddings, which are used as additional edge features of the input graph. An MPNN is then applied to this augmented graph. Specifically, let $h_v^t$ be the node representation for $v \in V$ in the $t$-th iteration. The update function is given by:

$$h_v^{t+1} = W_1^t(h_v^t, \sum_{u \in N(v)} W_2^t(h_u^t, h_v^t, e_{uv}, s_{uv}))$$ (4)

where $s_{uv}$ is the structural embedding for edge $uv$.

**Choice of structural embedding.** Recall the proof of Theorem 4.4 and the statement in Section 4 imply that the distance information within subgraphs is key to boosting the counting power of subgraph GNNs. Therefore, we encode such distance information, as well as some degree information, which provides the information of isomorphic types, as follows. An example is shown in Figure 1.

- The degree encoding: for each subgraph, we first compute the degree of all nodes within the subgraph, and then use the degree histogram as the encoding. For example, the degree histogram for the first subgraph (the subgraph rooted at edge $v_1v_3$) of Figure 1 is $(0, 0, 4, 0)$, since there are 4 nodes with degree 2 in the subgraph.

- The node-level distance encoding: for the subgraph rooted at edge $uv$, we use the shortest path distance histograms to the rooted nodes as the distance encoding. Take the first subgraph of Figure 1 as an example. The distance histogram for $v_1$ is $(1, 2, 1, 0)$, since there are one node with distance 0 ($v_1$), two nodes with distance 1 ($v_2$ and $v_3$), and one node with distance 2 ($v_4$) to node $v_1$. The same holds for the other rooted node $v_3$.

- The edge-level distance encoding: for the subgraph rooted at edge $uv$, define the label for each node as its shortest path distances to all the rooted nodes, i.e., $\forall u_1, f(u_1) = (d(u_1, u), d(u_1, v))$. Then we can define the label of edges in the subgraph as the concatenation of the label of its two end nodes, e.g., for edge $u_1v_1$, its label is $f(u_1v_1) = (f(u_1), f(v_1))$. We then use the edge-level distance histogram as the distance encoding. For example, for the subgraphs shown in Figure 1, there are seven types of edges: (0,1,1,0), (0,1,1,1), (0,1,1,2), (1,0,1,1), (1,0,2,1), (1,2,2,1), (2,1,2,1). Therefore the edge-level distance encoding for the first subgraph is (1,0,1,0,1,1,0), since there are one edge ($v_1v_3$) with label (0,1,1,0), one edge ($v_1v_2$) with label (0,1,1,2), one edge ($v_3v_4$) with label (1,0,2,1), and one edge ($v_2v_4$) with label (1,2,2,1).

Finally, we concatenate the three encodings to get the final structural embedding $s_{uv}$.

**Analysis on the structural embedding.** In terms of representation power, since all these distance encodings can be extracted using an MPNN within the subgraph, we conclude that:

**Proposition 5.1.** *ESC-GNN is not more powerful than subgraph GNNs rooted at 2-tuples with MPNN as the backbone GNN.*

In terms of algorithm efficiency, we can precompute these structural embeddings during subgraph extraction (the preprocessing cost can be amortized into each epoch/iteration), and ESC-GNN only needs to run the backbone GNN on the input graph. Therefore in every iteration, its computational

cost is only $O(|E|)$, and its memory cost is $O(|V|)$, both the same as MPNN. As for subgraph MPNNs rooted at edges, they need to run the process of subgraph extraction and run the backbone GNN over all subgraphs. In every iteration, their computational cost is $O(|E||E'_s|)$, and their memory cost is $O(|E||V'_s|)$, where $|V'_s|$ and $|E'_s|$ are the average numbers of nodes and edges among all subgraphs. Even if considering subgraph MPNNs rooted at nodes, in every iteration, their computational cost is $O(|V||E_s|)$, and their memory cost is $O(|V||V_s|)$. Note that $|V||E_s| >> \frac{|V|D}{2} = |E|$, where $D$ is the average node degree. Therefore we can safely conclude that ESC-GNN is more efficient than subgraph GNNs. We will empirically evaluate its efficiency in the experiment.

## 5.2 REPRESENTATION POWER OF ESC-GNN

In this section, we analyze the representation power of ESC-GNN from two perspectives: its counting power and its ability to distinguish non-isomorphic graphs.

**Counting power of ESC-GNN.** Existing works (You et al., 2021; Huang et al., 2023) mainly focus on subgraph counting. Here we provide results on both subgraph counting and induced subgraph counting. We use four popular types of substructures: cycles, cliques, stars, and paths, as examples to show the counting power of ESC-GNN. The proof is provided in the supplementary material.

**Theorem 5.2.** *In terms of subgraph counting, ESC-GNN can count (1) up to 4-cycles; (2) up to 4-cliques; (3) stars with arbitrary sizes; (4) up to 3-paths.*

**Theorem 5.3.** *In terms of induced subgraph counting, ESC-GNN can count (1) up to 4-cycles; (2) up to 4-cliques; (3) up to 4-stars; (4) up to 3-paths.*

**Compared with subgraph GNNs.** As shown in Proposition 5.1, ESC-GNN is less powerful than subgraph MPNNs rooted at 2-tuples (Huang et al., 2023). As for subgraph MPNNs rooted at nodes (Zhang & Li, 2021; You et al., 2021), they can only count up to 4-cycles, 3-cliques, and 3-paths (Huang et al., 2023). Therefore, ESC-GNN is more powerful than subgraph MPNNs rooted at nodes in terms of counting these substructures.

**The ability to distinguish non-isomorphic graphs.**

**Theorem 5.4.** *ESC-GNN is strictly more powerful than 2-WL, while not less expressive than 3-WL.*

**Compared with subgraph GNNs.** In terms of distinguishing non-isomorphic graphs, subgraph MPNNs rooted at nodes (Zhang & Li, 2021; You et al., 2021) are strictly less powerful than 3-WL (Frasca et al., 2022), while ESC-GNN is not less expressive than 3-WL.

In addition, although able to distinguish most pairs of non-isomorphic graphs (Babai et al., 1980), 2-WL fails to distinguish any pairs of non-isomorphic $r$-regular graphs with equal size. In this paper, we prove that ESC-GNN can distinguish almost all $r$-regular graphs:

**Theorem 5.5.** *Consider all pairs of $r$-regular graphs with $n$ nodes, let $3 \leq r < (2log2n)^{1/2}$ and $\epsilon$ be a fixed constant. With the hop parameter $h$ set to $\lfloor (1/2 + \epsilon)\frac{log2n}{log(r-1)} \rfloor$, there exists an ESC-GNN that can distinguish $1 - o(n^{-1/2})$ such pairs of graphs.*

## 6 EXPERIMENT

To thoroughly analyze the property of ESC-GNN, we evaluate it from the following perspectives: (1) we evaluate its representation power in Section 6.1, to show whether it can reach the theoretical power shown in Section 5.2; (2) we evaluate its performance on real-world benchmarks in Section 6.2, to show whether the increased representation power can boost its performance on real-world tasks; (3) we evaluate its efficiency in Section 6.3. The code is available at `https://anonymous.4open.science/r/ESC-GNN-D0E6`. We provide the experimental details, the analysis on the limitation of the paper, and the information of the used assets in the supplementary material.

**Baselines.** We compare with baseline methods including (1) a basic MPNN (Xu et al., 2018; Kipf & Welling, 2017); (2) subgraph GNNs including NGNN (Zhang & Li, 2021), IDGNN (You et al., 2021), GIN-AK+ (Zhao et al., 2022), SUN (Frasca et al., 2022), DSS-GNN (Bevilacqua et al., 2022), OSAN (Qian et al., 2022), and I$^2$-GNN (Huang et al., 2023); (3) high-order GNN models including 1-2-3-GNN (Morris et al., 2019) and PPGN (Maron et al., 2019); and (4) graph transformers including Graphormer-GD (Zhang et al., 2023) and GraphGPS (Rampášek et al., 2022).

Table 1: Evaluation on Counting Substructures (norm MAE), cells with MAE less than 0.01 (an indicator of successful counting (Huang et al., 2023)) are colored yellow.

| Dataset | Tailed Triangle | Chordal Cycle | 4-Clique | 4-Path | Triangle-Rectangle | 3-cycles | 4-cycles | 5-cycles | 6-cycles |
|---|---|---|---|---|---|---|---|---|---|
| MPNN | 0.3631 | 0.3114 | 0.1645 | 0.1592 | 0.2979 | 0.3515 | 0.2742 | 0.2088 | 0.1555 |
| ID-GNN | 0.1053 | 0.0454 | 0.0026 | 0.0273 | 0.0628 | 0.0006 | 0.0022 | 0.0490 | 0.0495 |
| NGNN | 0.1044 | 0.0392 | 0.0045 | 0.0244 | 0.0729 | 0.0003 | 0.0013 | 0.0402 | 0.0439 |
| GIN-AK+ | 0.0043 | 0.0112 | 0.0049 | 0.0075 | 0.1311 | 0.0004 | 0.0041 | 0.0133 | 0.0238 |
| PPGN | 0.0026 | 0.0015 | 0.1646 | 0.0041 | 0.0144 | 0.0005 | 0.0013 | 0.0044 | 0.0079 |
| $I^2$-GNN | 0.0011 | 0.0010 | 0.0003 | 0.0041 | 0.0013 | 0.0003 | 0.0016 | 0.0028 | 0.0082 |
| Graphormer-GD | 0.3660 | 0.2611 | 0.1580 | 0.1125 | 0.2460 | 0.3080 | 0.2317 | 0.1540 | 0.1380 |
| GraphGPS | 0.0132 | 0.0630 | 0.1156 | 0.0910 | 0.0551 | 0.0882 | 0.1645 | 0.0462 | 0.1193 |
| ESC-GNN | 0.0052 | 0.0169 | 0.0064 | 0.0254 | 0.0178 | 0.0074 | 0.0044 | 0.0356 | 0.0337 |

## 6.1 Representation Power of ESC-GNN

**Datasets.** We evaluate the representation power of ESC-GNN from two perspectives:

(a) Its ability to differentiate non-isomorphic graphs. We use (1) EXP (Abboud et al., 2021), which contains 600 pairs of non-isomorphic graphs that 1-WL/2-WL fails to distinguish; (2) SR25 (Balcilar et al., 2021), which contains 150 pairs of non-isomorphic strongly regular graphs that cannot be differentiated by 3-WL; (3) CSL (Murphy et al., 2019), which contains 150 regular graphs that 1-WL/2-WL fails to distinguish. These graphs are classified into 10 isomorphism classes. Classification accuracy is adopted as the evaluation metric.

(b) Its counting ability. We use the synthetic dataset from (Zhao et al., 2022). The task is to predict the number of substructures that pass each node in the given graph. The Mean Absolute Error (MAE) is adopted as the evaluation metric.

**Results.** In Table 2, ESC-GNN achieves 100% accuracy on all datasets. Considering that models as powerful as 3-WL (PPGN and 3-GNN) fail the SR25 dataset, the results serve as empirical evidence that ESC-GNN can effectively differentiate regular graphs (Theorem 5.5), and not less powerful than 3-WL (Theorem 5.4). In Table 1, ESC-GNN reaches less-than-0.01 MAE in terms of counting tailed triangles, 4-cliques, 3-cycles, and 4-cycles. The low error allows us to simply apply a rounding function to obtain the ground truth integer counting results, thus are considered an indicator of successful counting as in previous works (Huang et al., 2023). These counting results exactly match Theorem 5.2. Generally speaking, ESC-GNN performs much better than MPNNs, and slightly beats or performs comparably with node-based subgraph GNNs such as ID-GNN, NGNN and GIN-AK+. Also, it performs inferior to subgraph GNNs rooted at edges ($I^2$-GNN). This serves as the empirical evidence for Proposition 5.1. In addition, graph transformers such as GraphGPS [4] and Graphormer-GD have shown impressive performance in previous studies, particularly on graph-level prediction tasks. However, they perform inferior to subgraph GNNs and our proposed model on the substructure-counting task. This observation suggests potential limitations in the applicability of these transformer-based models to tasks involving substructure analysis.

Table 2: Test Accuracy on EXP/SR25/CSL

| Dataset | EXP | SR25 | CSL |
|---|---|---|---|
| MPNN | 50 | 6.67 | 10 |
| NGNN | 100 | 6.67 | - |
| GIN-AK+ | 100 | 6.67 | - |
| PPGN | 100 | 6.67 | - |
| 3-GNN | 99.7 | 6.67 | 95.7 |
| $I^2$-GNN | 100 | 100 | 100 |
| ESC-GNN | 100 | 100 | 100 |

Table 3: Evaluation on Algorithm Efficiency (Seconds).

| Dataset | ogbg-hiv | | ZINC | |
|---|---|---|---|---|
| Model | Pre | Run | Pre | Run |
| MPNN | 2.7 | 6296.8 | 6.2 | 1945.0 |
| NGNN | 1288.0 | 14862.9 | 300.3 | 8368.8 |
| $I^2$-GNN | 2806.5 | 1042963.7 | 677.7 | 18607.5 |
| GIN-AK+(Sample) | 376.2 | 10275 | 31.3 | 8862.9 |
| ESC-GNN | 1782.5 | 6301.0 | 362.4 | 2872.2 |

---

[4]We exclude the laplacian-based structural embeddings since they are not permutation-invariant, i.e., they may produce different outputs for the same graph.

Table 4: Evaluation on QM9 (MAE)

| Dataset | 1-GNN | 1-2-3-GNN | NGNN | $I^2$-GNN | ESC-GNN |
|---|---|---|---|---|---|
| $\mu$ | 0.493 | 0.476 | 0.428 | 0.428 | **0.231** |
| $\alpha$ | 0.78 | 0.27 | 0.29 | **0.230** | 0.265 |
| $\epsilon_{homo}$ | 0.00321 | 0.00337 | 0.00265 | 0.00261 | **0.00221** |
| $\epsilon_{lumo}$ | 0.00355 | 0.00351 | 0.00297 | 0.00267 | **0.00204** |
| $\Delta_\epsilon$ | 0.0049 | 0.0048 | 0.0038 | 0.0038 | **0.0032** |
| $R^2$ | 34.1 | 22.9 | 20.5 | 18.64 | **7.28** |
| ZPVE | 0.00124 | 0.00019 | 0.0002 | **0.00014** | 0.00033 |
| $U_0$ | 2.32 | 0.0427 | 0.295 | **0.211** | 0.645 |
| $U$ | 2.08 | **0.111** | 0.361 | 0.206 | 0.380 |
| $H$ | 2.23 | **0.0419** | 0.305 | 0.269 | 0.427 |
| $G$ | 1.94 | **0.0469** | 0.489 | 0.261 | 0.384 |
| $C_v$ | 0.27 | 0.0944 | 0.174 | **0.0730** | 0.105 |

## 6.2 REAL WORLD TASKS.

We present the results on QM9 (Ramakrishnan et al., 2014; Wu et al., 2018) in Table 4, and refer the readers to the supplementary material for additional real-world experiments. QM9 contains 130k small molecules, and the task is to perform regression on twelve graph properties. Graph transformers are not included since they need more than 3 days to predict one property.

Generally speaking, ESC-GNN performs better than classic MPNNs and slightly better than NGNN. This demonstrates that the proposed structural embedding can effectively extract valuable information from subgraph GNNs which benefits downstream tasks. We are surprised to find that in certain situations, we beat subgraph GNNs rooted at 2-tuples. This may be due to the fact that the framework of ESC-GNN is simple enough to avoid problems such as overfitting. We also observe that our performance on $U_0$, $U$, and $H$ in Table 4 is not good enough. These targets represent the Internal energy at 0K, the Internal energy at 298.15K, and the Enthalpy at 298.15K, respectively. To calculate these targets, computational methods take into account the interactions between all the atoms in the molecule and their surroundings, including any heat or work exchanged with the environment. As a result, globally expressive models (1-2-3 GNN) can achieve the best performance, while subgraph GNNs, which utilize local information to enhance graph representation, perform much worse.

## 6.3 ALGORITHM EFFICIENCY.

In terms of algorithm efficiency, we compare ESC-GNN with four baselines: an MPNN, NGNN, and GIN-AK+ whose subgraphs are rooted at node, and $I^2$-GNN whose subgraphs are rooted at 2-tuples. We accelerate the code of $I^2$-GNN by implementing a parallel subgraph preprocessing strategy. We report the data preprocessing time and the standard running time (100 epochs for ogbg-hiv, and 1000 epochs for ZINC) in Table 3. We point out that although sampling strategies used in GIN-AK+ make it faster, they lose the theoretical substructure counting ability as the substructures appearing in unsampled subgraphs will not be counted. Therefore, it is not fair to compare it with our approach. Nonetheless, to provide a comprehensive analysis, we have included a comparison in Table 3. As shown in the table, ESC-GNN exhibits a notable advantage in terms of total running time compared to node-based subgraph GNNs, even when employing sampling methods lacking theoretical guarantees. This is because the preprocessing needs only to be done once and can be readily used in all the later training and inference stages. This is consistent with our observation in Section 5.1.

## 7 CONCLUSION

The huge computational cost is associated with subgraph GNNs due to the requirement of running backbone GNNs among all subgraphs. To address this challenge and enable efficient substructure counting with GNNs, we theoretically show that the distance information within subgraphs is key to boosting the counting power of GNNs. We then encode this information into a structural embedding and enhance standard GNN models with this embedding, eliminating the need to learn representations over all subgraphs. Experiments on various benchmarks demonstrate that the proposed model retains the representation power of subgraph GNNs while running much faster. It can potentially enhance the utility of subgraph GNNs in a variety of applications that require efficient substructure counting.

**Reproducibility Statement**

The code is available at `https://anonymous.4open.science/r/ESC-GNN-D0E6`.

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
