# AN EFFICIENT SUBGRAPH GNN WITH PROVABLE SUBSTRUCTURE COUNTING POWER
# – SUPPLEMENTARY MATERIAL –

In the supplementary material, we provide:

1. the introduction of the Weisfeiler-Leman algorithm (WL) and Message Passing Neural Network (MPNN);

2. the proof of Theorem 3.2 in the main paper (the upper bound of $m$-WL in terms of counting substructures);

3. the proof of Theorem 3.4 in the main paper (the lower bound of subgraph GNN in terms of counting connected substructures)

4. the proof of Theorem 4.4 in the main paper (ESC-GNN's ability to differentiate non-isomorphic graphs);

5. the proof of Theorem 4.2 and Theorem 4.3 in the main paper (ESC-GNN's ability for subgraph counting and induced subgraph counting);

6. the proof of Theorem 4.5 (ESC-GNN's ability to differentiate regular graphs);

7. additional discussion on the expressive power of ESC-GNN;

8. experimental details;

9. additional experiments on real-world benchmarks and ablation study;

10. limitations and the assets we used.

## 1 THE WEISFEILER-LEMAN ALGORITHM AND MESSAGE PASSING NEURAL NETWORK

**1-WL.** We first describe the classic Weisfeiler-Leman algorithm (1-WL). For a given graph $G$, 1-WL aims to compute the coloring for each node in $V$. It computes the node coloring for each node by aggregating the color information from its neighborhood iteratively. In the 0-th iteration, the color for node $v \in V$ is $c_v^0$, denoting its initial isomorphic type[1]. For labeled graphs, the isomorphic type of a node is simply its node feature. For unlabeled graphs, we give the same 1 to all nodes. In the $t$-th iteration, the coloring for $v$ is computed as

$$c_v^t = \text{HASH}(c_v^{t-1}, \{\!\!\{c_u^{t-1} : u \in N(v)\}\!\!\})$$

, where HASH is a bijective hashing function that maps the input to a specific color. The process ends when the colors for all nodes between two iterations are unchanged. If two graphs have different coloring histograms in the end (e.g., different numbers of nodes with the same color), then 1-WL detects them as non-isomorphic.

**$k$-WL.** For each $k \geq 2, k \in \mathbb{N}$, the $k$-dimensional WL algorithm ($k$-WL) colors $k$-tuples instead of nodes. In the 0-th iteration, the isomorphic type of a $k$-tuple $\vec{v}$ is given by the hashing of 1) the tuple of colors associated with the nodes of the $\vec{v}$, and 2) the adjacency matrix of the subgraph induced by $\vec{v}$ ordered by the node ordering within $\vec{v}$. In the $t$-th iteration, its coloring is updated by:

$$c_{\vec{v}}^t = \text{HASH}(c_{\vec{v}}^{t-1}, c_{\vec{v},(1)}^t, ..., c_{\vec{v},(k)}^t)$$

---

[1]The term "isomorphic type" is based on previous work (Morris et al., 2019), which gives each $k$-tuple of nodes an initial feature such that two $k$-tuples receive the same initial feature if and only if their induced subgraphs (indexed by node order in the $k$-tuple) are isomorphic.

, where

$$c_{\vec{v},(i)}^t = \{\!\!\{c_{\vec{u}}^{t-1} | \vec{u} \in N_i(\vec{v})\}\!\!\}, i \in [k]$$

. Here, $N_i(\vec{v}) = \{(v_1, ..., v_{i-1}, w, v_{i+1}, ..., v_k) | w \in V\}$ is the $i$-th neighborhood of $\vec{v}$. Intuitively, $N_i(\vec{v})$ is obtained by replacing the $i$-th component of $\vec{v}$ by each node from $V$. Besides the updating function, other procedures of $k$-WL are analogous to 1-WL. In terms of distinguishing graphs, 2-WL is as powerful as 1-WL, and for $k \geq 2$, $(k+1)$-WL is strictly more powerful than $k$-WL.

**MPNNs.** MPNNs are a class of GNNs that learns node representations by iteratively aggregating messages from neighboring nodes. Let $h_v^t$ be the node representation for $v \in V$ in the $t$-th iteration. It is usually initialized with the node's intrinsic attributes. In the $(t+1)$-th iteration, it is updated by

$$h_v^{t+1} = W_1^t(h_v^t, \sum_{u \in N(v)} W_2^t(h_u^t, h_v^t, e_{uv}))$$

, where $W_1^t$ and $W_2^t$ are two learnable functions. MPNN's power in terms of distinguishing non-isomorphic graphs is upper bounded by 2-WL (Xu et al., 2018).

## 2 THE PROOF OF THEOREM 3.2

We begin by introducing the *graph isomorphism*. For a pair of graphs $G_1 = (V_1, E_1)$ and $G_2 = (V_2, E_2)$, if there exists a bijective mapping $f : V_1 \rightarrow V_2$, so that for any edge $(u_1, v_1) \in E_1$, it satisfies that $(f(u_1), f(v_1)) = (u_2, v_2) \in E_2$, then $G_1$ is isomorphic to $G_2$, otherwise they are not isomorphic. Up to now, there is no polynomial algorithm for solving the graph isomorphism problem. One popular method is to use the $k$-order Weisfeiler-Leman (Weisfeiler & Leman, 1968) algorithm ($k$-WL). It is known that 1-WL is as powerful as 2-WL, and for $k \geq 2$, $(k+1)$-WL is more powerful than $k$-WL.

We then prove that $k$-WL can't count all connected substructures with $(k+1)$ nodes (specifically, $(k+1)$-cliques). We restate the result as follows:

**Theorem 2.1.** *For any $k \geq 2$, there exists a pair of graphs $G$ and $H$, such that $G$ contains a $(k+1)$-clique as its subgraph while $H$ does not, and that $k$-WL can't distinguish $G$ from $H$.*

*Proof.* The counter-example is inspired by the well-known Cai-Fürer-Immerman (CFI) graphs (Cai et al., 1992). We define a sequence of graphs $G_k^{(\ell)}, \ell = 0, 1, \ldots, k+1$ as following,

$$
\begin{aligned}
V_{G_k^{(\ell)}} = \Big\{ u_{a,\vec{v}} \Big| a \in [k+1], \vec{v} \in \{0,1\}^k \text{ and} \\
\left. \begin{array}{ll} \vec{v} \text{ contains an even number of 1's,} & \text{if } a = 1, 2, \ldots, k - \ell + 1, \\ \vec{v} \text{ contains an odd number of 1's,} & \text{if } a = k - \ell + 2, \ldots, k + 1. \end{array} \right\} 
\end{aligned}
\tag{1}
$$

Two nodes $u_{a,\vec{v}}$ and $u_{a',\vec{v}'}$ of $G_k^{(\ell)}$ are connected iff there exists $m \in [k]$ such that $a' \mod (k+1) = (a+m) \mod (k+1)$ and $v_m = v'_{k-m+1}$. We have the following lemma.

**Lemma 2.2.** *(a) For each $\ell = 0, 1, \ldots, k+1$, $G_k^{(\ell)}$ is an undirected graph with $(k+1)2^{k-1}$ nodes;*

*(b) The set of graphs $G_k^{(\ell)}$ with an odd $\ell$ are mutually isomorphic; similarly, the set of graphs $G_k^{(\ell)}$ with an even $\ell$ are mutually isomorphic.*

It's easy to verify (a). To prove (b), it suffices to prove $G_k^{(\ell)}$ is isomorphic to $G_k^{(\ell+2)}$ for all $\ell = 0, 1, \ldots, k-1$. We apply a renaming to the nodes of $G_k^{(\ell)}$: we flip the $1^{\text{st}}$ bit of $\vec{v}$ in every node named $u_{k-\ell,\vec{v}}$, and flip the $k^{\text{th}}$ bit of $\vec{v}$ in every node named $u_{k-\ell+1,\vec{v}}$. Since this is a mere renaming of nodes, the resulting graph is isomorphic to $G_k^{(\ell)}$. However, it's also easy to see that the resulting graph follows the construction of $G_k^{(\ell+2)}$. Therefore, we assert that $G_k^{(\ell)}$ must be isomorphic to $G_k^{(\ell+2)}$.

Now, let's ask $G = G_k^{(0)}$ and $H = G_k^{(1)}$. Obviously there is a $(k+1)$-clique in $G$: nodes $u_{j,0^k}, j = 1, 2, \ldots, k+1$ are mutually adjacent by definition of $G_k^{(0)}$. On the contrary, we have

**Lemma 2.3.** *There's no $(k+1)$-clique in $H$.*

The proof is given below. Assume there is a $(k+1)$-clique in $H$. Since there's no edge between nodes $u_{a,\vec{v}}$ with an identical $a$, the $(k+1)$-clique must contain exactly one node from every node set $\{u_{a,\vec{v}}\}$ for each fixed $a \in [k+1]$. We further assume that the $(k+1)$ nodes are $u_{a,b_{a1}b_{a2}...b_{ak}}, a = 1, 2, \ldots, k+1$. Using the condition for adjacency, we have

$$b_{2k} = b_{11}, \tag{2}$$

$$b_{3k} = b_{21}, b_{3(k-1)} = b_{12}, \tag{3}$$

$$b_{4k} = b_{31}, b_{4(k-1)} = b_{22}, b_{4(k-2)} = b_{13}, \tag{4}$$

$$\cdots\cdots\cdots\cdots$$

$$b_{(k+1)k} = b_{k1}, b_{(k+1)(k-1)} = b_{(k-1)2}, \ldots, b_{(k+1)1} = b_{1k}. \tag{5}$$

Applying the above identities to the summation

$$\sum_{a=1}^{k+1}\sum_{j=1}^{k} b_{aj} = 2 \sum_{j=1}^{k} \left( b_{1j} + b_{2j} + \cdots + b_{(k-j+1)j} \right), \tag{6}$$

we see that it should be even. However, by definition of $G_k^{(1)}$, there are an even number of 1's in $b_{a1}b_{a2}\ldots b_{ak}$ when $a \in [k]$, and an odd number of 1's when $a = k+1$. Therefore, the sum in equation 6 should be odd. This leads to a contradiction.

Finally, to prove the $k$-WL equivalence of $G$ and $H$, we have

**Lemma 2.4.** *$k$-WL can't distinguish $G$ and $H$.*

By virtue of the equivalence between $k$-WL and pebble games (Grohe & Otto, 2015), it suffices to prove that Player II will win the $\mathcal{C}_k$ bijective pebble game on $G$ and $H$. We state the winning strategy for Player II as following. Since $G$ and $H$ are isomorphic with nodes $\{u_{k+1,*}\}$ deleted, Player II can always choose an isomorphism $f : G - \{u_{k+1,*}\} \to H - \{u_{k+1,*}\}$ to survive if Player I never places a pebble on nodes $u_{k+1,*}$. Furthermore, since $k$ pebbles can occupy nodes with at most $k$ different values of $a$ (in $u_{a,\vec{v}}$), there's always a set of pebbleless nodes $\{u_{a_0,\vec{v}}\}$ with some $a_0 \in [k+1]$. Therefore, Player II only needs to do proper renaming on $H$ between $u_{k+1,*}$ and $u_{a_0,*}$ as stated above. This makes every $\vec{v}$ in $u_{a_0,\vec{v}}$ have an odd number of 1's. Player II then chooses the isomorphism on $G - \{u_{a_0,*}\}$ and $H^{\text{renamed}} - \{u_{a_0,*}\}$. This way, Player II never loses since there are not enough pebbles for Player I to make use of the oddity at the currently pebbleless set of nodes.

$\square$

*Remark* 2.5. Notice that if we take $k = 2$, then $G$ is two 3-cycles while $H$ is a 6-cycle, which 2-WL cannot differentiate; if we take $k = 3$, then $G$ is the 4*4 Rook's graph while $H$ is the Shrikhande graph, which 3-WL cannot differentiate. In these special cases, the above construction complies with our well-known examples.

## 3 THE PROOF OF THEOREM 3.4

We restate the theorem as follows:

**Theorem 3.1.** *For any connected substructure with no more than $k + m$ ($m \geq 2, k > 0$) nodes, there exists a subgraph GNN rooted at $k$-tuples with backbone GNN as powerful as $m$-WL that can count it.*

*Proof.* Based on Remark 3.3 in the main paper, there exists a type of connected $k$-tuple that satisfies the decomposition of the connected substructure. Then the substructure can be separated into 2 subgraphs: the nodes that belong to the $k$-tuple (we call them rooted nodes) and the nodes that do not belong to the $k$-tuple (we call them non-rooted nodes). Formally, for the given two substructures $G_1 = (V_1, E_1)$ and $G_2 = (V_2, E_2)$, we define the subgraph that formed by the rooted nodes of $G_1$ ($G_2$, resp.) as $G_{1,r} = (V_{1,r}, E_{1,r})$ ($G_{2,r} = (V_{2,r}, E_{2,r})$, resp.). Similarly, define the subgraph that formed by the non-rooted nodes of $G_1$ ($G_2$, resp.) as $G_{1,n} = (V_{1,n}, E_{1,n})$ ($G_{2,n} = (V_{2,n}, E_{2,n})$,

resp.). We can also define the subgraph formed between the rooted nodes and non-rooted nodes of $G_1$ ($G_2$, resp.) as $G_{1,c} = (V_{1,c}, E_{1,c})$ ($G_{2,c} = (V_{2,c}, E_{2,c})$, resp.). Then it is easy to find that for $G_1$, $V_1 = (V_{1,r} \cup V_{1,n})$, $V_{1,c} \subseteq V_1$, $E_1 = E_{1,r} \cup E_{1,n} \cup E_{1,c}$. There is no intersection between $V_{1,r}$ and $V_{1,n}$, and there is no intersection between $E_{1,r}$, $E_{1,c}$, and $E_{1,n}$. The same holds for $G_2$.

If $G_1$ and $G_2$ are non-isomorphic, then there can be three potential situations: (1) $G_{1,r}$ and $G_{2,r}$ are non-isomorphic; (2) $G_{1,n}$ and $G_{2,n}$ are non-isomorphic; (3) $G_{1,c}$ and $G_{2,c}$ are non-isomorphic. In the following section, we will prove that in all these situations, the subgraph GNN can distinguish between $G_1$ and $G_2$.

$G_{1,r}$ **and $G_{2,r}$ are non-isomorphic.** It denotes that for $G_1$ and $G_2$, the selected $k$-tuples are different. Then of course the subgraph GNN can differentiate $G_1$ and $G_2$.

$G_{1,n}$ **and $G_{2,n}$ are non-isomorphic.** Recall that we use a backbone GNN as powerful as $m$-WL to encode the information within the subgraph, and $V_{1,n}$ and $V_{2,n}$ contain nodes no more than $m$ nodes. Therefore if $G_{1,n}$ and $G_{2,n}$ are non-isomorphic, then they will have different isomorphic types, and thus can be distinguished by the backbone GNN.

$G_{1,c}$ **and $G_{2,c}$ are non-isomorphic.** We define the label of all nodes in $G_1$ (the same holds for $G_2$) as its distance to the nodes in the $k$-tuple. Formally, let $V_{1,r} = \{v_{1,r,1}, ..., v_{1,r,k}\}$, then $\forall u \in V_1$, its label $f_1(u) = (d(u, v_{1,r,1}), ..., d(u, v_{1,r,k}), I(u))$, where $d(.)$ denotes the shortest path distance between two nodes, and $I(u)$ denotes the label that reflects the isomorphic type of $u$ encoded by the subgraph GNN within $G_{1,n}$.

Since the substructures are connected, there exists at least a node $u_1 \in V_{1,n}$, whose label contains at least an index with the value "1". For $f(u_1)$, the indices with value "1" denotes that there exist edges between $u_1$ and the corresponding nodes in $V_{1,r}$. While the indices with value larger than 1 denote that there is no edge between $u_1$ and the corresponding nodes in $V_{1,r}$. The same holds for $u_2$ and $V_{2,r}$. Therefore, if the subgraph formed by $u_1$ and $V_{1,r}$ and the subgraph formed by $u_2$ and $V_{2,r}$ are non-isomorphic, then $f_1(u_1)$ and $f_2(u_2)$ are different, and the subgraph GNN can differentiate the two substructures. Also, if the $I(u_1)$ and $I(u_2)$ are different, then the subgraph GNN can also distinguish $G_1$ and $G_2$. We can then consider the next nodes, and continue the process inductively.

Therefore, if for all nodes in $V_{1,n}$, we can find a unique node in $V_{2,n}$ that has the same label as it. Then $G_{1,c}$ and $G_{2,c}$ are isomorphic. Reversely, if $G_{1,c}$ and $G_{2,c}$ are non-isomorphic, then there exists at least a node in $V_{1,n}$, that we cannot find a unique node in $V_{2,n}$ that has the same label as it. Therefore the subgraph GNN can differentiate $G_1$ and $G_2$.

Then we can assign each substructure a unique color according to its isomorphic type, and use the color histogram of the graph as the output function. If two graphs have different numbers of certain substructures, then the color histogram will be different.

Based on the above results, the subgraph GNN can count the given type of connected substructure.

$\square$

## 4 The Proof of Theorem 4.4

We state the result as follows.

**Theorem 4.1.** *ESC-GNN is strictly more powerful than 2-WL, while not less expressive than 3-WL.*

*Proof.* **ESC-GNN is not less powerful than 3-WL.** As shown in Theorem 5.1, ESC-GNN is able to count 4-cliques. In the pair of graphs called the 4*4 Rook Graph and the Shrikhande Graph (shown in Figure 1), there exist several 4-cliques in the 4*4 Rook Graph, while there is no 4-clique in the Shrikhande Graph. Therefore ESC-GNN can differentiate the pair of graphs. Considering that 3-WL cannot differentiate them (Arvind et al., 2020), ESC-GNN is not less powerful than 3-WL.

**ESC-GNN is more powerful than 2-WL.** Using the MPNN (Xu et al., 2018) as the backbone network, it can be as powerful as 2-WL in terms of distinguishing non-isomorphic graphs. However, there exist pairs of graphs, e.g., the 4*4 Rook Graph and the Shrikhande Graph that can be distinguished by ESC-GNN but not 2-WL. Therefore, ESC-GNN is strictly more powerful than the 2-WL.

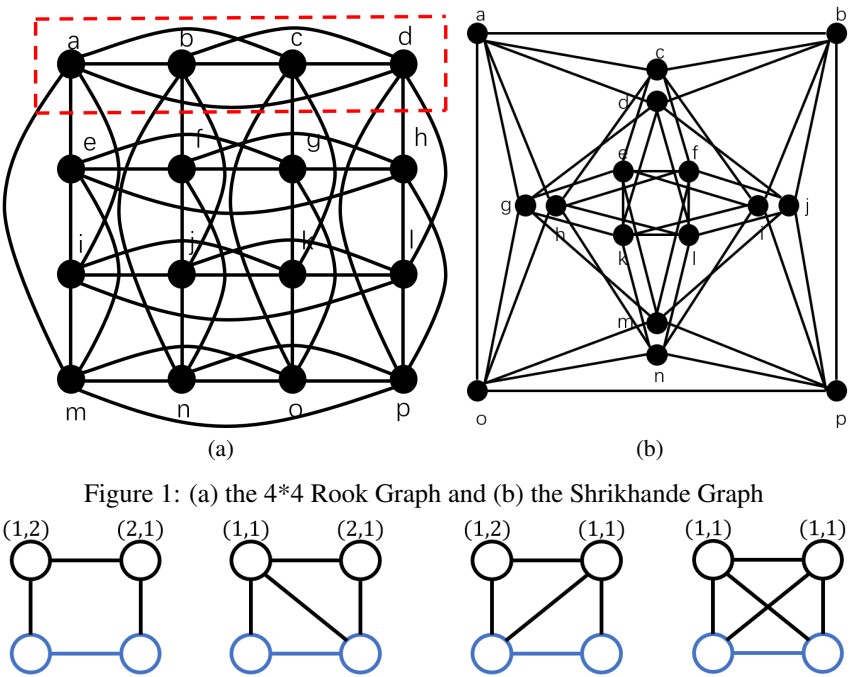

Figure 1: (a) the 4*4 Rook Graph and (b) the Shrikhande Graph

Figure 2: Examples of 4-cycles that pass the rooted edges. In these figures, the rooted 2-tuples are colored blue.

□

## 5 THE PROOF OF THEOREM 4.2

Below we show the counting power of ESC-GNN in terms of subgraph counting.

**Theorem 5.1.** *In terms of subgraph counting, ESC-GNN can count (1) up to 4-cycles; (2) up to 4-cliques; (3) stars with arbitrary sizes; (4) up to 3-paths.*

*Proof.* **Clique counting.** The number of 3-cliques is the number of nodes with the shortest path distance "1" to both rooted nodes; the number of 4-cliques is the number of edges labeled $(1, 1, 1, 1)$ (definition see the edge-level distance encoding in Section 4 in the main paper, and example see Figure 2(d)). Therefore, ESC-GNN can count these types of cliques. In terms of 5-cliques, 4-WL cannot count them according to Theorem 2.1, therefore subgraph GNNs rooted on edges with MPNN as the backbone GNN cannot count 5-cliques according to Proposition 3.5 in the main paper. Then according to Proposition 4.1 in the main paper, ESC-GNN also cannot count 5-cliques.

**Cycle counting.** the counting of 3-cycles is the same as 3-cliques. In terms of counting 4-cycles, there are basically 4 different situations where 4-cycles exist, examples are shown in Figure 2. Note that figures (a),(b), and (c) contain one 4-cycle, respectively, while figure (d) contains two 4-cycles that pass the rooted edge. Therefore the number of 4-cycles is the weighted sum of the number of edges with labels $(1, 2, 2, 1)$, $(1, 1, 2, 1)$, $(1, 2, 1, 1)$, $(1, 1, 1, 1)$.

In terms of 5-cycle subgraph-counting, we provide a counter-example in Figure 3. In Figure 3(a), there is one 5-cycle $ABEDC$ that passes $AB$, while there is no 5-cycle that passes $AB$ in Figure 3(b). Considering that the degree information and the distance information is the same for the pair of graphs, ESC-GNN cannot differentiate them. Therefore, ESC-GNN cannot subgraph-count 5-cycles.

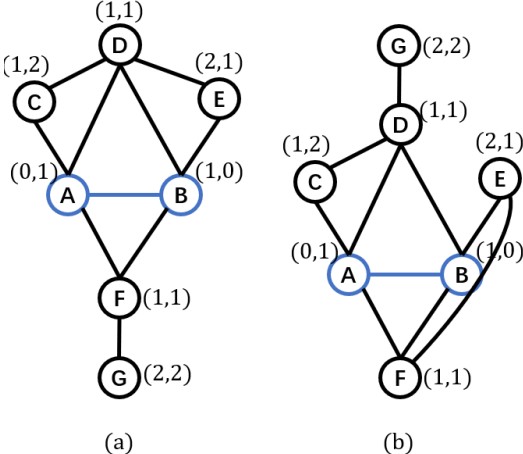

Figure 3: Examples where ESC-GNN cannot subgraph-count 5-cycles. In these figures, the rooted 2-tuples are colored blue.

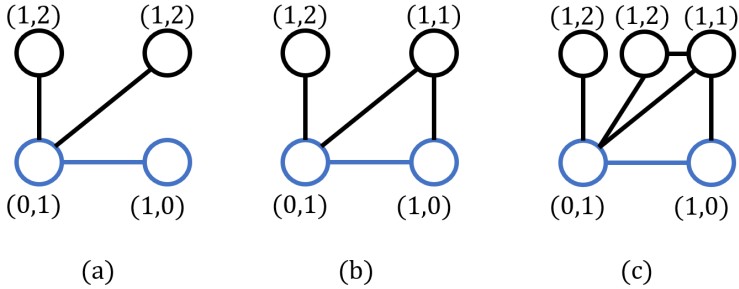

Figure 4: Examples of stars that pass the rooted edges. In these figures, the rooted 2-tuples are colored blue.

**Path Counting.** Note that the definition of paths here is different from other substructures. It denotes the number of edges within the path instead of the number of nodes within the path. Here, we slightly extend the use of 2-tuples in ESC-GNN by considering not only 2-tuples with edges but also 2-tuples without edges.

In terms of counting 2-paths or 3-paths between 2 nodes, it is equal to counting 3-cycles or 4-cycles, between edges, respectively. We have proven that ESC-GNN can count these cycles, therefore ESC-GNN can count such edges.

**Star counting.** We can decompose the graph-level star counting problem to 2-tuples by considering the first node of each 2-tuple as the root of stars. Examples are shown in Figure 4. We advocate that the number of stars is easily encoded by the number of nodes whose shortest path distance is 1 to the first rooted node. Denote the number of nodes with the shortest path distance 1 to the first rooted node as $N'$ (including the second node), then the number of $p$-stars is $C_{N'-1}^{p-2}$. A similar proof is provided by (Chen et al., 2020).

$\square$

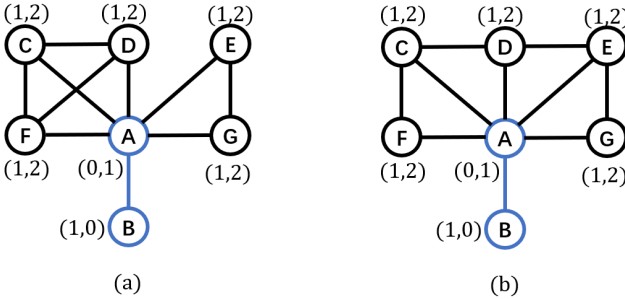

Figure 5: Examples where ESC-GNN cannot induced-subgraph-count 5-stars. In these figures, the rooted 2-tuples are colored blue.

## 6 THE PROOF OF THEOREM 4.3

Below we show the counting power of ESC-GNN in terms of induced subgraph counting. We restate the Theorem 4.3 as follows.

**Theorem 6.1.** *In terms of induced subgraph counting, ESC-GNN can count (1) up to 4-cycles; (2) up to 4-cliques; (3) up to 4-stars; (4) up to 3-paths.*

*Proof.* **Clieque counting.** Since cliques are fully connected substructures, the proof of cliques is the same as the proof of subgraph counting.

**Cycle counting.** For cycles, the number of 3-cycles is the same as 3-cliques. As for 4-cycles, we only need to consider the situation shown in Figure 2(a), where the number of $(1, 2, 2, 1)$ edges reflects the number of 4-cycles. For induced-subgraph-counting 5-cycles, in Figure 6(a), there is no 5-cycle that pass $AB$, while in Figure 6(b), there is one 5-cycle $ABEDC$ that passes $AB$. However, ESC-GNN cannot differentiate the two graphs since the degree information and the distance information is the same. Therefore, ESC-GNN cannot induced-subgraph-count 5-cycles. It can serve as the same example for not counting 4-paths.

**Path counting.** The proof of paths is actually the same as Theorem 5.1.

**Star counting.** In terms of 3-stars and 4-stars, we only need to consider the situation shown in Figure 4(a), where the number of $(1, 2)$ nodes encodes the number of stars, i.e., denote the number of $(1, 2)$ nodes as $N'$, the number of $p$-stars ($p \leq 4$) is $C_{N'}^{p-2}$.

For 5-stars, a pair of examples are shown in Figure 5. These two graphs will be assigned the same structural embedding since the node degree information, and the distance information among these two graphs are the same. Therefore, ESC-GNN cannot differentiate the two graphs. However, Figure 5(a) contains no 5-star that passes the rooted 2-tuple $AB$, while Figure 5(b) contains one 5-star ($BAFDG$) that passes the rooted 2-tuple $AB$. ☐

## 7 THE PROOF OF THEOREM 4.5

Here we restate Theorem 4.5 as follows:

**Theorem 7.1.** *Consider all pairs of $r$-regular graphs with $n$ nodes, let $3 \leq r < (2log2n)^{1/2}$ and $\epsilon$ be a fixed constant. With the hop parameter $h$ set to $\lfloor (1/2 + \epsilon) \frac{log2n}{log(r-1)} \rfloor$, there exists an ESC-GNN that can distinguish $1 - o(n^{-1/2})$ such pairs of graphs.*

Recall that we encode the distance information as augmented structural features for ESC-GNN. For the input graph $G$, denote the node-level distance encoding on $h$-hop subgraphs on 2-tuple $(u, v)$ as $D_{(u,v),G}^{h,\text{node}}$. Note that we can naturally transfer the encoding on 2-tuples to node by: ($D_{v,G}^{h,\text{node}} = D_{(v,v),G}^{h,\text{node}}$). We then introduce the following lemma.

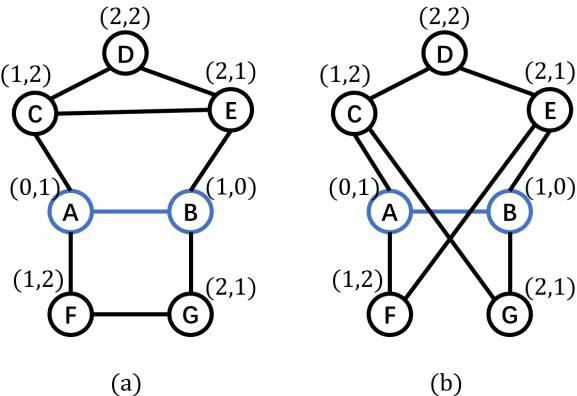

Figure 6: Examples where ESC-GNN cannot induced-subgraph-count 5-cycles. In these figures, the rooted 2-tuples are colored blue.

**Lemma 7.2.** *For two graphs $G_1 = (V_1, E_1)$ and $G_2 = (V^2, E^2)$ that are randomly and independently sampled from $r$-regular graphs with n nodes ($3 \le r < (2log2n)^{1/2}$). Select two nodes $v_1$ and $v_2$ from $G_1$ and $G_2$ respectively. Let $h = \lfloor (1/2 + \epsilon)\frac{log2n}{log(r-1)} \rfloor$ where $\epsilon$ is a fixed constant, $D^{h,node}_{v_1,G_1} = D^{h,node}_{v_2,G_2}$ with probability at most $o(n^{-3/2})$.*

*Proof.* The proof follows from (Bollobás, 1982; Feng et al., 2022). As $D^{h,node}_{v,G}$ stores exactly the same information as the node configuration used in (Feng et al., 2022), therefore the proof is exactly the same as the proof of Lemma 1 in (Feng et al., 2022).

Based on Lemma 7.2, we can prove Theorem 7.1. Given node $v_1 \in V_1$, compare $D^{h,node}_{v_1,G_1}$ with each $D^{h,node}_{v_2,G_2}$ where $v_2 \in V_2$. The probability that $D^{h,node}_{v_1,G_1} \ne D^{h,node}_{v_2,G_2}$ for all possible $v_2 \in V_2$ is $1 - o(n^{-3/2}) * n = 1 - o(n^{-1/2})$. Therefore, ESC-GNN can distinguish $1 - o(n^{-1/2})$ such pairs of graphs. □

## 8   ADDITIONAL DISCUSSION ON THE EXPRESSIVE POWER OF ESC-GNN

In previous sections, we mainly discuss the expressive power enhanced by the proposed structural encoding. In this section, we shift to the integration of the global message passing layer and elucidate its substantial contribution to the expressive power. This is exemplified below through a toy example: consider graph $G$, composed of a 4-cycle and a 1-path (edge), and graph $H$, comprising a 5-path (a path with 6 nodes). When extracting 1-hop subgraphs for each node, both $G$ and $H$ yield two 1-paths and four "V graphs" (3 nodes and 2 edges). Consequently, the subgraph representation fails to distinguish between these two graphs. However, the introduction of a global message passing layer enables differentiation due to the distinct connections among these subgraphs. This distinction is applicable to both subgraph GNNs and our framework. A more comprehensive analysis can be found in (Rattan & Seppelt, 2023), which substantiates that the global message passing can enhance expressiveness regardless of the choice of the hop parameter.

Additionally, We conduct an ablation study on the substructure counting dataset by removing the global message passing layer from ESC-GNN. The resulting new model, denoted as "(-MP)", relies solely on aggregating the structural encoding to derive the final prediction. Given the absence of additional node attributes, the new model can be easily implemented as ESC-GNN with a single message passing layer that aggregates the edge-level structural embedding to obtain the node-level prediction. The result is presented in Table 1. As evident from the table, ESC-GNN consistently outperforms the new model across all benchmarks, demonstrating that the global message passing layer significantly boosts the representation power of ESC-GNN.

Table 1: Ablation study on the proposed structural embedding (norm MAE).

| Dataset | Tailed Triangle | Chordal Cycle | 4-Clique | 4-Path | Triangle-Rectangle | 3-cycles | 4-cycles | 5-cycles | 6-cycles |
|---|---|---|---|---|---|---|---|---|---|
| MPNN | 0.3631 | 0.3114 | 0.1645 | 0.1592 | 0.2979 | 0.3515 | 0.2742 | 0.2088 | 0.1555 |
| ESC-GNN | 0.0052 | 0.0169 | 0.0064 | 0.0254 | 0.0178 | 0.0074 | 0.0044 | 0.0356 | 0.0337 |
| (- MP) | 0.062 | 1.9355 | 0.4124 | 0.0271 | 0.0563 | 0.1508 | 0.0050 | 0.0996 | 0.0443 |

Table 2: Statistics of the used datasets.

| Dataset | Graphs | Avg Nodes | Avg Edges | Task Type | Metric |
|---|---|---|---|---|---|
| MUTAG | 188 | 17.9 | 19.8 | Graph Classification | ACC |
| PTC-MR | 349 | 14.1 | 14.5 | Graph Classification | ACC |
| ENZYMES | 600 | 32.6 | 62.1 | Graph Classification | ACC |
| PROTEINS | 1113 | 39.1 | 72.8 | Graph Classification | ACC |
| IMDB-BINARY | 1000 | 19.8 | 96.5 | Graph Classification | ACC |
| ZINC-12k | 12000 | 23.2 | 24.9 | Graph Regression | MAE |
| ogbg-molhiv | 41127 | 25.5 | 27.5 | Graph Classification | AUC-ROC |
| ogbg-molpcba | 437929 | 26.0 | 28.1 | Graph Classification | AP |
| QM9 | 129433 | 18.0 | 18.6 | Graph Regression | MAE |
| Synthetic | 5000 | 18.8 | 31.3 | Node Regression | MAE |

## 9 EXPERIMENTAL DETAILS

**Stastics of Datasets.** The statistics of all used datasets are available in Table 2.

**Experimental details.** The baselines and data splittings of our experiments follow from existing works (Zhang & Li, 2021; Huang et al., 2023) and the standard data split setting. For ESC-GNN, we adopt GIN (Xu et al., 2018) as the backbone GNN. In the structural embedding, we use both the shortest path distance and the resistance distance (Lü & Zhou, 2011) as the distance feature. For the hop parameter $h$ we search between 1 to 4, and report the best results. Following existing works (Huang et al., 2023), we use Adam optimizer as the optimizer, and use plateau scheduler with patience 10 and decay factor 0.9. On most datasets, the learning rate is set to 0.001, and the hidden embedding dimension is set to 300. The training epoch is set to 2000 for counting substructures, 400 for QM9, 1000 for ZINC, and 150 for the OGB dataset. Most of the experiments are implemented with two Intel Core i9-7960X processors and 2 NVIDIA 3090 graphics cards. Others (e.g., experiments on the TU dataset) are implemented with two Intel Xeon Gold 5218 processors and 10 NVIDIA 2080TI graphics cards.

## 10 EVALUATION ON REAL-WORLD DATASETS

**Molecule Dataset.** We evaluate ESC-GNN on various popular real-world molecule datasets, including ZINC (Dwivedi et al., 2020) and the OGB dataset (Hu et al., 2020). ZINC is a dataset of chemical compounds, and the task is graph regression. For the OGB dataset, we use ogbg-molhiv and ogbg-molpcba for evaluation. ogbg-molhiv contains 41K molecules with 2 classes, and ogbg-molpcba contains 438 molecules with 128 classes. The task is to predict to which these molecules belong. We follow the standard evaluation metric and the dataset split, and report the result in Table 3.

**TU datasets.** We evaluate the performance of ESC-GNN on the TU datasets (Morris et al., 2020). The experimental settings follow (Zhang & Li, 2021) for more consistent evaluation standards. Specifically, we uniformly use the 10-fold cross validation framework, with the split ratio of training/validation/test set 0.8/0.1/0.1. The results are available in Table 4. In the table, ESC-GNN (h2) and ESC-GNN (h3) denote ESC-GNN with the hop parameters setting to 2 and 3.

Table 3: Evalation on ZINC and OGB datasets.

| Dataset | OGBG-HIV (AUCROC) | OGBG-PCBA (AP) | ZINC |
|---|---|---|---|
| GIN (Xu et al., 2018) | 77.07±1.49 | 27.03±0.23 | 0.163 |
| PNA (Corso et al., 2020) | 79.05±1.32 | 28.38±0.35 | 0.188 |
| GSN (Bouritsas et al., 2022) | 77.99±0.01 | - | 0.115 |
| DGN (Beaini et al., 2021) | 79.70±0.97 | 28.85±0.30 | 0.168 |
| CIN (Bodnar et al., 2021) | **80.94±0.57** | - | **0.079** |
| NGNN (Zhang & Li, 2021) | 78.34±1.86 | 28.32±0.41 | 0.111 |
| GIN-AK+ (Zhao et al., 2022) | 79.61±1.19 | **29.30±0.44** | 0.080 |
| OSAN (Qian et al., 2022) | - | - | 0.126 |
| SUN (Frasca et al., 2022) | 80.03±0.55 | - | 0.083 |
| DSS-GNN (Bevilacqua et al., 2022) | 76.78±1.66 | - | 0.102 |
| I$^2$-GNN (Huang et al., 2023) | 78.68±0.93 | - | 0.083 |
| Graph Transformer (Rampášek et al., 2022) | 77.40±1.77 | 27.51±0.28 | 0.113 |
| ESC-GNN | 78.62±1.06 | 28.16±0.31 | 0.086 |

## 10.1 RESULTS

**Comparison with backbone models.** On OGBG-MolHIV, OGBG-MolPCBA, and the TU dataset, we use GIN (Xu et al., 2018) as the backbone model. While on ZINC, we use a plain graph transformer without positional encodings from (Rampášek et al., 2022) as the backbone model. The experimental results compared with these backbones demonstrate that the proposed structural information not only enhances the theoretical representation power but also improves the empirical representation power.

**Comparison with subgraph GNNs.** When compared to subgraph GNNs rooted at 2-tuples, such as I$^2$-GNN, ESC-GNN performs slightly worse or comparably on these benchmark tasks. This empirical observation provides evidence that the proposed structural embedding effectively captures valuable information from subgraph GNNs, benefiting downstream tasks. It is worth noting that I$^2$-GNN generally outperforms node-based subgraph GNNs like NGNN and GIN-AK+, suggesting that the theoretical advantages gained from subgraph selection and aggregation policies contribute to the empirical representation power. Furthermore, among the subgraph GNNs, those that introduce interactions between subgraphs, such as SUN, GIN-AK+, and OSAN, tend to perform better compared to subgraph GNNs that do not incorporate such interactions, like NGNN and ID-GNN. This observation indicates that subgraph interaction also plays a role in enhancing the representation power of subgraph GNNs. Exploring additional techniques to incorporate this interaction can be considered for future research.

**Comparison with GNNs that incorporate the substructure information.** GSN (Bouritsas et al., 2022) incorporates the number of specific substructures as augmented features. In contrast to GSN, our framework does not explicitly encode the number of substructures. Instead, we encode the more general distance information of subgraphs, which enables plain GNNs to count many substructures without the need to manually determine which substructures to include. Regarding the real-world dataset, GSN achieves a MAE score of 0.115 on ZINC, whereas ESC-GNN attains a MAE score of 0.096. Similarly, utilizing the same backbone (GIN), GSN records an AUCROC score of 77.99 on OGBG-HIV, whereas ESC-GNN demonstrates a higher AUCROC score of 78.62. These outcomes underscore the superiority of our proposed model.

## 11 ABLATION STUDY

We evaluate the effectiveness of each part of the proposed structural embedding on the substructure counting dataset. For the proposed three types of structural embedding (the degree encoding, the node-level distance encoding, and the edge-level distance encoding), we delete one of them from the original structural embedding every time and report the results in Table 5. We observe that after removing the three types of embedding, ESC-GNN performs worse compared with its original version, especially after removing the edge-level distance encoding. This is consistent with our theoretical results: in Theorem 3.1, we show that the proposed structural embedding contains key information for the counting power of subgraph GNNs; in Theorem 5.1 and Theorem 6.1, we show that the edge-level distance information directly encodes the number of certain types of substructures.

Table 4: Experiments on TU, Accuracy as the evaluation metric.

| Dataset | MUTAG | PTC-MR | PROTEINS | ENZYMES | IMDB-B |
|---|---|---|---|---|---|
| GIN | 84.5±8.9 | 51.2±9.2 | 70.6±4.3 | 38.3±6.4 | 73.3±4.7 |
| PPGN | 84.7±8.2 | 55.0±6.4 | 74.8±3.3 | 55.0±6.4 | 71.5±5.4 |
| NGNN | 87.9±8.2 | 54.1±7.7 | 73.9±5.1 | 29.0±8.0 | 73.1±5.7 |
| GIN-AK+ | **88.8±4.0** | 60.5±8.0 | 75.5±4.4 | **58.9±6.2** | 72.4±3.7 |
| SUN | 86.1±6.0 | 60.2±7.2 | 72.1±3.8 | 16.7±0.0 | **73.7±2.9** |
| $I^2$-GNN | 87.9±4.3 | **61.4±8.7** | 74.8±2.9 | 40.3±6.7 | 73.6±4.0 |
| ESC-GNN (h2) | 86.2±7.9 | 52.9±6.4 | 73.3±4.1 | 53.2±8.1 | 72.0±6.0 |
| ESC-GNN (h3) | 85.6±7.9 | 56.4±6.9 | **76.0±4.5** | 43.3±6.0 | **73.7±4.8** |

Table 5: Ablation study on the proposed structural embedding (norm MAE).

| Dataset | Tailed Triangle | Chordal Cycle | 4-Clique | 4-Path | Triangle-Rectangle | 3-cycles | 4-cycles | 5-cycles | 6-cycles |
|---|---|---|---|---|---|---|---|---|---|
| MPNN | 0.3631 | 0.3114 | 0.1645 | 0.1592 | 0.2979 | 0.3515 | 0.2742 | 0.2088 | 0.1555 |
| ESC-GNN | 0.0052 | 0.0169 | 0.0064 | 0.0254 | 0.0178 | 0.0074 | 0.0044 | 0.0356 | 0.0337 |
| (- degree) | 0.0121 | 0.0492 | 0.0106 | 0.0322 | 0.0349 | 0.0342 | 0.0144 | 0.0513 | 0.0652 |
| (- node-level dist) | 0.0382 | 0.0344 | 0.0222 | 0.0428 | 0.0256 | 0.0157 | 0.0261 | 0.0492 | 0.0608 |
| (- edge-level dist) | 0.0208 | 0.2811 | 0.0497 | 0.0584 | 0.185 | 0.2617 | 0.2244 | 0.1654 | 0.1364 |

## 12 EVALUATION ON THE SPACE COST

With regards to the preprocessing time, our approach's preprocessing time is comparable to that of $I^2$-GNN as shown in Table 3 in the main paper since we use their code to extract subgraph information. Although we require a small amount of extra time to extract and preprocess the structural embeddings, we believe that this is a reasonable trade-off since the actual model running time is reduced by orders. When compared to the total running time of subgraph GNNs, the preprocessing time is negligible. Therefore, our ESC-GNN's total running time is less than 1% of that of $I^2$-GNN on ogbg-hiv.

As for the storage of structural embeddings, we only need to store a vector of integer indices for each structural embedding, as illustrated in Figure 1 in the main paper, without really storing any dense high-dimensional vectors. Therefore, the storage is still manageable. Specifically, when feeding the indices to the backbone model, we use a learnable matrix (which is a model parameter) to transform the integer index vector into dense embeddings. For example, to extract the degree information of the first subgraph in Figure 1 in the main paper, we use a learnable matrix $W \in \mathbb{R}^{4 \times h}$ and compute the degree information with $W * [0; 0; 4; 0]$, where $*$ denotes sparse matrix multiplication (which is very fast for sparse matrices), and the sparse vector $[0; 0; 4; 0]$ is what we need to store. This approach requires relatively small storage space.

We have also included the space cost of our approach in Table 6. In these datasets, ESC-GNN requires much less storage space than $I^2$-GNN while slightly more space than NGNN. These results demonstrate that our approach for storing structural embeddings offers excellent storage performance.

Table 6: Evaluation on the space cost.

| Dataset | OGBG-HIV | ZINC | QM9 |
|---|---|---|---|
| Original | 159MB | 16.2MB | 281MB |
| NGNN | 2.32GB | 218MB | 2.90GB |
| $I^2$-GNN | 5.95GB | 627MB | 8.25GB |
| ESC-GNN | 2.57GB | 427MB | 4.46GB |

Table 7: Evaluation on Counting Substructures on ZINC (norm MAE).

| Tasks | 3-cycle | 4-cycle | 5-cycle | 6-cycle |
|---|---|---|---|---|
| GNN | 0.0016 | 0.0030 | 0.0394 | 0.1442 |
| GIN-AK+ | 0.0009 | 0.0064 | 0.0036 | 0.0057 |
| NGNN | 0.0002 | 0.0001 | 0.0007 | 0.0012 |
| $I^2$-GNN | 0.0003 | 0.0001 | 0.0003 | 0.0007 |
| ESC-GNN | 0.0008 | 0.0004 | 0.0058 | 0.0047 |

## 13 COUNTING SUBSTRUCTURES ON OTHER DATASETS

In order to evaluate the counting power of the proposed model, we conducted experiments on the ZINC dataset by generating the cycle counting task and reporting the MAE result in Table 7. To count cycles, we used the simple-cycle function from `https://networkx.org/documentation/stable/reference/algorithms/generated/networkx.algorithms.cycles.simple_cycles.html`. Surprisingly, we found that all methods achieved much better MAE scores than those reported in Table 1 of the main paper. This may be due to the fact that many graphs in ZINC contain few cycles, resulting in a small MAE value.

In general, the reported results are consistent with Table 1 of the main paper. ESC-GNN performs better in counting 3-cycles and 4-cycles, and performs worse on 5-cycles and 6-cycles. This observation is consistent with Theorem 5.1. Furthermore, it outperforms MPNN, performs comparably to GIN-AK+, and is outperformed by $I^2$-GNN, which is consistent with our theoretical results, showing that the proposed structural embedding can extract valuable information from the subgraph GNNs.

## 14 LIMITATIONS AND THE ASSETS WE USED

**Limitations of the paper.** First, we have shown that the representation power of our model is bounded by 4-WLs and subgraph GNNs rooted on 2-tuples in terms of distinguishing non-isomorphic graphs and counting substructures.

Second, the proposed model may not reach a satisfying performance on benchmarks where the encoded substructures are of no use. Also, the proposed model may not suit high-order graphs where the neighbors of the nodes and edges are defined differently from the simple graphs.

**The assets we used.** Our model is experimented on benchmarks from (Dwivedi et al., 2020; Hu et al., 2020; Zhao et al., 2022; Morris et al., 2020; Abboud et al., 2021; Balcilar et al., 2021; Murphy et al., 2019; Ramakrishnan et al., 2014; Wu et al., 2018) under the MIT license.