# OpenReview forum: "An Efficient Subgraph GNN with Provable Substructure Counting Power"
_ICLR.cc/2024/Conference — Submitted to ICLR 2024_

### Official Review · Reviewer_aWFU · 2023-10-30

**Soundness:** 3 good
**Presentation:** 3 good
**Contribution:** 2 fair
**Rating:** 6
**Confidence:** 4

**Summary:**

This paper proposes a new technique for efficient substructure counting (ESC) for graph neural networks (GNNs), called ESC-GNN. To this end, the paper deeply explores existing subgraph GNN methods in the literature and establishes expressiveness result linked to the WL hierarchy for these. The paper then shows that using sub-graphs offers an efficiency gain, as the higher-order WL test can run on (induced) sub-graphs, shifting a high polynomial weight to sets of smaller graphs and only keeping a $k^th$ polynomial power over the input graph size related to the number of connected $k-$tuples being considered.

Building on this result, the paper describes a structural encoding method for edges in an input graph that occurs at pre-computation: ESC-GNN considers all rooted subgraphs around connected 2-tuples, i.e., edges, and computes a structural encoding using node degree and edge distance information. The paper then shows how this approach, which is substantially more efficient than running an MPNN on each subgraph separately, allows to detect important substructures (4-cycles, 4-cycles, etc.) and how this approach is strictly more powerful than 2-WL (folklore 1-WL), while not being less powerful than 3-WL (folklore 2-WL). Finally, the paper conducts a large set of synthetic experiments to validate the strength of their model, as well as experiments on real-world benchmarks (results on QM9, efficiency on OGBG-hiv, ZINC), demonstrating the speed and good performance of their approach.

**Strengths:**

- Presenting an efficient pre-computation to mitigate the complexity of sampling and running MPNNs over subgraphs is a well-justified and well-thought contribution.

- The presentation is clear: I particularly enjoyed the running example in Figure 1, as this really helped me follow along with the structure encoding computation. The theoretical results in the background are also well-presented.

- The synthetic experiments show the value of the approach, particularly in light of its efficiency.

- The experimental analysis of QM9 results, particularly on tasks where ESC-GNN performs less well, is thorough.

**Weaknesses:**

- The experimental results are not very compelling. In particular, real-world experimental results for QM9 are strong in part, but performance on other tasks is substantially worse. Moreover, results on OGBG-HIV, ZINC (reported in the appendix), which ideally should also be in the main paper given that efficiency is reported on these, are not strong. Just to be clear: it is completely acceptable not to achieve SOTA results across the board, particularly against specialized approaches. However, I do expect some analysis of results (just like in QM9), leading to real-world scenarios where ESC-GNN is a clear and obvious candidate for use and would achieve the best results. This comment also applies to the synthetic experiments. I therefore ask the authors to revisit their experimental section and modify their analysis towards establishing a well-defined use case for their approach.

- The paper's approach to subgraph counting is specialized towards common patterns, and only establishes relatively simple results. This is not a major weakness in itself (and is common in subgraph GNN literature), particularly given the complexity of general (induced) subgraph counting. However, the paper would be more interesting / compelling if it were to discuss more general sub-structure detection (more explicitly than via a connection to k-WL).  To this point, the authors can strengthen the work by conducting case studies on real-world datasets to establish the importance of detecting cycles/paths/cliques. This would nicely complement the existing ablation studies in the appendix, and provide a meaningful explanation of when ESC-GNN is useful. As it stands, a main concern with subgraph GNNs is their over-specialization to pre-defined graph structures, and so any results / experiments to show more general structure detection / strong performance beyond the pre-designed use case, would substantially strengthen this paper.

All in all, I think this paper has a place in the subgraph GNN literature, as it is well-written and offers an efficient solution for counting (almost) all the same small subgraphs. However, the weaknesses I raise above prevent me from more strongly supporting this work. Nonetheless, I am happy to revise my rating should the authors address my concerns.

**Questions:**

No direct questions. Please address the weaknesses / suggestions I provide in the weaknesses section above.

---

> ### Author Response · Authors · 2023-11-18
> **Response to Reviewer aWFU (1/2)**
>
> Thank you for your valuable feedback and insightful suggestions. Your comments have been diligently considered, and we will integrate your suggestions into our revised manuscript.
>
>
> **Importance of substructures.** We have incorporated an ablation study to show the importance of substructures in real-world tasks. The substructures, as detailed in Table 1, include 3-6 cycles, tailed triangles, chordal cycles, 4-cliques, 4-paths, and triangle-rectangles. We first report the number of these substructures at graph-level and node-level. Specifically, we report the average number of these substructures per graph (denoted as "graph") and the average number of these substructures that pass each node (denoted as "node"). The statistics are shown below.
>
> |OGBG-HIV |3-cycle | 4-cycle | 5-cycle | 6-cycle | tailed triangle | chordal cycle | 4-clique | 4-paths | triangle-rectangles |
> |  :----: |  :----: | :----: | :----: | :----: | :----: | :----: | :----: | :----: | :----: |
> |graph |  2.69e-2 |  3.69e-2 | 7.00e-1 | 2.28 | 1.02e-1 | 4.84e-3 | 2.43e-5 | 68.38 | 9.36e-3|
> | node |  3.16e-3 | 5.79e-3 | 1.37e-1 | 5.37e-1 | 4.01e-3 | 3.79e-4 | 7.63e-6 | 5.40 | 3.67e-4 |
>
> |ZINC |3-cycle | 4-cycle | 5-cycle | 6-cycle | tailed triangle | chordal cycle | 4-clique | 4-paths | triangle-rectangles |
> |  :----: |  :----: | :----: | :----: | :----: | :----: | :----: | :----: | :----: | :----: |
> |graph |  6.41e-2  | 1.60e-2 | 8.52e-1 | 1.80 | 8.79e-2 | 0 | 0 |  55.15 | 0 |
> | node | 8.30e-3 | 2.76e-3 | 1.84e-1 | 4.67e-1 | 3.80e-3 | 0 | 0 | 4.76 | 0 |
>
>
> Note that 6-cycles are commonly observed within these two graphs. On average, a graph contains approximately two 6-cycles, and there is an average incidence of 0.5 6-cycles that pass a node. This observation underscores the significance of 6-cycle, whose presence may potentially be attributed to the existence of benzene rings.
>
> We then evaluate the significance of these substructures. In particular, we use a base GIN framework as the backbone network, and adopt the count of substructures that pass each node as the augmented node features. In the table, “all” denotes adding all these substructures as the augmented node feature. Due to the time limit, we only run the experiment twice and report the mean result.
>
> | Dataset | Baseline |3-cycle | 4-cycle | 5-cycle | 6-cycle | tailed triangle | chordal cycle | 4-clique | 4-paths | triangle-rectangles | all |
> |  :----: |  :----: |  :----: | :----: | :----: | :----: | :----: | :----: | :----: | :----: | :----: | :----: |
> | ZINC |0.149| 0.145 | 0.150 | 0.143 | 0.115 | 0.146 | 0.152 | 0.149| 0.167 |0.149 | 0.100 |
> | OGBG-HIV |73.89| 73.48 | 74.67 | 74.47 | 73.60 | 73.69  | 71.98 | 73.56| 68.40| 53.18| 71.47 |
>
> **Analysis on the real-world benchmarks.**
>
> We observe that in most cases, the number of substructures does not boost the performance either on ZINC or on OGBG-HIV. The only exception is that the number of 6-cycles boosts the performance on ZINC (maybe due to the information of benzene rings), and the number of all these substructures boosts the performance on ZINC. We present potential illustrations for the observation:
>
> Firstly, in these molecule graphs, it is imperative to incorporate not only structural but also semantic information to enhance the model. For instance, it is well-established that molecular fingerprints can significantly elevate performance on the OGBG-HIV dataset (Refer: https://ogb.stanford.edu/docs/leader_graphprop/#ogbg-molhiv). In particular, the fingerprints contain the number of specific molecule structures, encompassing both the graph structures of molecules and the semantic information of atom types and bond types.
>
> Secondly, the general distance information contains more structural information than the number of specific substructures, thus contributing more to real-world performance. For instance, we have proven that the number of 4-cycles can be computed using the distance information. Conversely, deducing distance information solely from the count of 4-cycles is impracticable, given the multiplicity of configurations that can form a 4-cycle.
>
> Based on the observation, we extract further information from the distance encoding by refining the original method. In our original implementation, we solely employ an embedding layer for the extraction of distance information. During the rebuttal stage, we integrate a 2-layer MLP, supplemented with two batch normalization layers and two ELU activation layers, subsequent to the embedding layer. As shown in the table below, the implementation of this enhanced model structure on both ZINC and OGBG-HIV datasets has yielded a significant amplification in performance, providing empirical evidence that the distance encoding contains valuable information which includes but is not limited to the number of substructures.
>
> |dataset |ZINC | OGBG-HIV |
> |  :----: |  :----: | :----: |
> | ESC-GNN| 0.086$\pm$0.003 | 78.62$\pm$1.06|
> | ESC-GNN (new) | 0.078$\pm$0.004 |79.21$\pm$0.84|

---

> > ### Author Response · Authors · 2023-11-18
> > **Response to Reviewer aWFU (2/2)**
> >
> > **Some comments on the proposed encoding.** Prior to delving into the analysis of general structure detection, it is important to point out that the proposed distance encoding is not specially designed for substructure counting. Rather, it should be considered as a catalyst for future research endeavors that develop an efficient (pre-computed) encoding scheme preserving the representation power (not limited to the counting power) of subgraph GNNs.
> >
> >
> >
> > **More general structure detection.** The number of substructures with $n$ nodes will grow exponentially with the increasing of $n$. In particular, each of the $n$ nodes can be potentially connected to any of the remaining $n-1$ nodes, yielding a total of $\frac{n(n-1)}{2}$ possible edges. Given that each edge either exists or does not exist, there are $2^{\frac{n(n-1)}{2}}$ possible substructures in total.
> >
> > This enumeration includes multiple instances of isomorphic substructures. The maximum occurrence of any isomorphism type within this enumeration does not exceed $n!$. Therefore, the count of distinct isomorphism types of substructures with $n$ nodes is at least $\frac{2^{\frac{n(n-1)}{2}}}{n!}$. As a consequence, considering only the popular substructures can be a more practical choice.
> >
> > Still, we can extend the theoretical results to a more general extent: **ESC-GNN can count any connected substructures with no more than 4 nodes.**
> >
> > These substructures include edges (with 2 nodes), 2-paths, 3-cycles, (with 3 nodes), 3-paths, 4-cycles, tailed triangles, 4-cliques, and the “cyclic graph with a diagonal” (denote as $C$) (with 4 nodes).
> >
> > It is obvious that ESC-GNN can count edges, and we have proven that ESC-GNN can count 2-paths, 3-cycles, 3-paths, 4-cycles, and 4-cliques. We now prove that ESC-GNN can count tailed triangles and $C$.
> >
> > **Counting of $C$.** Assume that the number of $(1, 1)$ nodes as $k$. Recall that the $(1,1)$ nodes denote the nodes whose distances to the two target nodes are both $1$. The number of $C$ that passes the two target nodes is $\frac{k(k-1)}{2}$
> >
> > **Counting of tailed triangles.** The tailed triangle is formed by a triangle and a tailed node, and there are generally two situations for the tailed node: (1) the tailed node is exclusively linked to a single target node; (2) the tailed node is connected to both target nodes. The latter scenario aligns identically with $C$, which can be computed by ESC-GNN. Addressing the first situation, assume the number of $(1,1)$ nodes is $k$, the number of $(1, 2)$ nodes is $m$, and the number of $(2,1)$ nodes is $n$. Then the number of situation one is exactly $k(m+n)$.

---

> > > ### Comment · Reviewer_aWFU · 2023-11-18
> > > **Reviewer Response**
> > >
> > > I thank the authors for their response, and for providing additional experimental results. Overall, I still have the same concerns about the strength of the approach and the significance of your theoretical analyses:
> > >
> > > - *On significance of substructures for real-world experiments.* It is still unclear where this approach would be practically strong, and where the benefit of this model lies beyond simply including relevant sub-graphs based on the target task, e.g., ZINC. The new model variation also does not fundamentally change the above point.
> > >
> > > - *On structure detection.* I understand the current capabilities of ESC-GNN. I was instead alluding to providing alternative analyses beyond finite-size structure detection and the k-WL connection, which mostly builds on existing work, to strengthen the paper's contribution.
> > >
> > > All in all, I will maintain my initial verdict.

---

> ### Author Response · Authors · 2023-11-19
>
> We are grateful for the time spent reading our comments, and deeply appreciate your suggestions.
>
> > On significance of substructures for real-world experiments. It is still unclear where this approach would be practically strong, and where the benefit of this model lies beyond simply including relevant sub-graphs based on the target task, e.g., ZINC. The new model variation also does not fundamentally change the above point.
>
> We note that the proposed framework demonstrates superior performance over all baseline methods on ZINC, whose prediction relies on local information such as the synthetic accessibility and cycles. Furthermore, the framework has also achieved the best result across various properties in the QM9 benchmark. This serves as empirical evidence underscoring the framework's capability to enhance performance in molecular benchmarks, particularly those that rely on local structural features for attribute prediction.
>
> As for OGBG-HIV, we note that the inhibition of HIV replication involves a complex interplay of both local attributes (such as the structure and function of specific molecules and their enzyme active sites) and global attributes (such as the overall graph structure of both the virus and the molecule components within cells and their intersections). Therefore, the SOTA methods integrate global information (such as graph pooling models or graph transformers) with local information (such as the fingerprint, which contains the count of various specific molecules and may possess a degree of domain-specificity).
>
>
> > On structure detection. I understand the current capabilities of ESC-GNN. I was instead alluding to providing alternative analyses beyond finite-size structure detection and the k-WL connection, which mostly builds on existing work, to strengthen the paper's contribution.
>
> We are pleased to extend our analyses to encompass additional structures. However, to ensure computational efficiency and focus, we kindly request that you specify particular types or classes of structures for analysis. As shown in the previous response, the exhaustive enumeration of all substructures presents a significant computational challenge and may not be feasible within the scope of this study.
>
> We sincerely thank you again for your insightful suggestions and constructive reviews.

---

### Official Review · Reviewer_p2Va · 2023-10-30

**Soundness:** 2 fair
**Presentation:** 2 fair
**Contribution:** 2 fair
**Rating:** 5
**Confidence:** 4

**Summary:**

The authors study the counting power of those subgraph GNNs that do not exchange information between subgraphs. More precisely, they show that these are expressive in the _graph-level_ counting of connected substructures. They then propose a framework named ESC-GNN that extracts subgraphs to compute distance features within each subgraph, and use them as structural embeddings of the original graph which is then processed through a GNN. Finally, the paper shows theoretical results on the counting power of the proposed framework.

**Strengths:**

The problem is interesting and the study of the counting powers of WL tests and subgraph GNNs is valuable on its own. The proposed architecture is simple but expressive for the task of substructure counting.

**Weaknesses:**

I think the major weakness is that __it seems that the model loses permutation equivariance due to the order of the encodings__ in $s_{uv}$. Specifically, consider the node-level distance encoding: for the subgraph rooted at edge $uv$ we have a distance histogram for $v$ and a distance histogram for $u$. Those two are concatenated (along with the other encodings) in $s_{uv}$. But which of the two distance histograms should be the first in the concatenation (that for $u$ or that for $v$)? If the edge is undirected no ordering should be preferred so choosing according to the node id leads to a choice that is not permutation equivariant.

The second main weakness is that __Proposition 4.5 does not seem to follow from previous work, and it is not proven in the paper__. I think that the claim on page 5: ``Previous works (Geerts, 2020; Frasca et al., 2022) show that for m ≥ 2 .. a subgraph GNN rooted at k-tuples with backbone GNN as powerful as m-WL can be implemented by (m + k)-IGN.'' is not true. Indeed it was shown only for $k=1$ and $m=2$. Therefore Proposition 4.5 is not immediate. I think it should be related to Proposition 2 in Qian et al 2022, which proves the same for any $k$ and $m=1$.

**Questions:**

1. Please expand on the order of the two node-level distance encoding, as well as on Proposition 4.5, as explained in the Weaknesses.
2. On page 7, the claim ``As shown in Proposition 5.1, ESC-GNN is less powerful than subgraph MPNNs rooted at 2-tuples" does not seem correct, as according to Proposition 5.1 they can be as powerful as subgraph MPNNs. Please clarify.
3. Does Theorem 4.4 hold for both induced and non-induced substructures?
4. The experimental section can be improved:

    a. Why do you focus on node-level tasks? I understand that node-level implies graph-level but the contrary is not true. Since you focus on graph-level tasks in the theoretical part, I don't understand why you test on node-level tasks. Furthermore, I noticed there is an additional counting experiments on ZINC in the appendix, but why is it limited to cycle counting?

    b. The time comparison on ZINC and OGB is presented in the main paper without reporting the results on those datasets in the main paper. Please move the results on these datasets in the main paper.

   c. Why results on ZINC do not include the std or average across seeds? And why do you use a graph transformers? What are the results with a GNN as a backbone model?

---

> ### Author Response · Authors · 2023-11-18
> **Response to Reviewer p2Va (1/2)**
>
> Thank you for your valuable feedback and insightful suggestions. Your comments have been diligently considered, and we will integrate your suggestions into our revised manuscript.
>
>
> > W1. I think the major weakness is that it seems that the model loses permutation equivariance due to the order of the encodings in
> . Specifically, consider the node-level distance encoding: for the subgraph rooted at edge $\dots$ If the edge is undirected no ordering should be preferred so choosing according to the node id leads to a choice that is not permutation equivariant.
>
> > Q1. Please expand on the order of the two node-level distance encoding, as well as on Proposition 4.5, as explained in the Weaknesses.
>
> **A1.** Actually, our model satisfies permutation equivariance (PE). For your question, the information for node $u$ will be first in the concatenation. This is because our distance encoding is based on $k$-tuples which rely on the sequence of nodes rather than undirected edges that do not preserve the node sequence. Therefore, $s_{uv}$ and $s_{vu}$ can be different since $(u,v)$ and $(v,u)$ are different 2-tuples.
>
> We provide an intuitive illustration of why ESC-GNN satisfies PE. When computing the node representation for $u$, ESC-GNN aggregates the information from its neighbors $\\{v | v \in N(u)\\}$. In this aggregation, the model sums over all $s_{uv}$ instead of $s_{vu}$. The approach ensures that the sequence remains consistent and unaffected by any permutations.
>
> > W2. The second main weakness is that Proposition 4.5 does not seem to follow from previous work, and it is not proven in the paper. I think that the claim on page 5 $\dots$ Therefore Proposition 4.5 is not immediate. I think it should be related to Proposition 2 in Qian et al 2022, which proves the same for any k and m=1
>
>
> **A2.** Thanks for your suggestion very much. We will add a formal proof in the revised manuscript, and provide a brief proof below. Huang et al. (2023) show that the distance information within subgraphs preserves the same representation power as the node marking policy, i.e., gives the rooted nodes an identity marking. We will then show that the subgraph GNN based on node marking policy can be implemented by $(k+m)$-IGN. (the word “implement” follows from Definition 3 in (Frasca et al.)).
>
> First, the node marking policy can be implemented by the $(k+m)$-IGN following the proof of Lemma 4 in (Frasca. et al.). Assume that for a graph $G$, its input to the $m$-IGN is a $n^m * c_1$ matrix $X$, where $n$ is the number of nodes, and $c_1$ is the feature dimension. For the corresponding subgraphs, the input to the $(k+m)$-IGN is a $n^{(m+k)} * c_2$ matrix $H$, where $c_2$ is the feature dimension. For the $k$-tuple $(v_1, v_2, \dots, v_k)$, we have $H[v_1, v_2, \dots, v_k, v_i, \dots, v_i] = X[v_i, \dots, v_i]$  $\bigoplus$  $\mathbb{1}_{i, k}$
> for each $i \in \\{1, \dots, m\\}$. In the equation,
>
> $\mathbb{1}_{i, k}$  denotes the one-hot $k$-dimensional vector being $1$ in dimension $i$ and 0 elsewhere and $\bigoplus$ denotes the concatenation between vectors. For other entries, $H[v_1, v_2, \dots, v_k, :, \dots, ] = X$ $\bigoplus$
>
> $\mathbb{0}_{k}$ (the space is left since markdown will produce a wrong formula without the space), where $\mathbb{0}_k$
>
> denotes a zero vector of dimension $k$. Since the subgraph GNN models the permutation equivariant operations within $n^{m+k}$, it can be implemented by the $(m+k)$-IGN. An example can be found in Lemma 5 in (Frasca et al.)). Specifically, in the message passing period of (Frasca et al.), the node representation $X_{iii}$ can be updated by $X_{iij}$ and $X_{ijj}$, the node representation $X_{iji}$ can be updated by $X_{iji}$ and $X_{iii}$, and $X_{ijk}$ can be represented by $X_{ijk}$ and $X_{ijj}$. In a more general situation, we can replace the first node $i$ with the $k$-tuple, and the latter two nodes with the $m$-tuple that can implement the operation of $m$-WL (Geerts, 2020). As for the message aggregation period, the subgraph pooling followed by a global pooling can be represented by a non-adjacent-to-adjacent pooling followed by a global summation. Therefore the subgraph GNN can be implemented by $(m+k)$-IGN, and thus is bounded by $(m+k)$-WL.
>
>
>
> > Q2. On page 7, the claim ``As shown in Proposition 5.1, ESC-GNN is less powerful than subgraph MPNNs rooted at 2-tuples" does not seem correct, as according to Proposition 5.1 they can be as powerful as subgraph MPNNs. Please clarify.
>
> **A3.** For example, subgraph MPNNs rooted at 2-tuple can count 5-cycles (Huang et al. 2023), while ESC-GNN cannot. Together with Proposition 5.1, ESC-GNN is less powerful than subgraph MPNNs rooted at 2-tuples. We will clarify it in the revised manuscript.

---

> ### Author Response · Authors · 2023-11-18
> **Response to Reviewer p2Va (2/2)**
>
> > Q3. Does Theorem 4.4 hold for both induced and non-induced substructures?
>
> **A4.** It holds both for induced subgraph counting and subgraph counting. We will clarify it in the paper.
>
>
> > Q4. The experimental section can be improved: a. Why do you focus on node-level tasks? I understand that node-level implies graph-level but the contrary is not true. Since you focus on graph-level tasks in the theoretical part, I don't understand why you test on node-level tasks.
>
> **A5.** The experiments follow the setting of (Huang et al.). We choose it instead of the graph-level counting task since it is more challenging, i.e., we can compute the number of substructures within the graph by the weighted sum of the number of substructures that pass each node. However, the reverse does not necessarily hold. Although our proof mostly focuses on graph-level counting, it can be potentially transferred to node-level counting. We briefly illustrate it below.
>
> Our proof has shown that the 2-tuple level counting of the mentioned substructures can be done. For a substructure $s$, assumes the number of $s$ that passes the 2-tuple $(u,v)$ as $n_{uv}$, and the number of automorphism of the 2-tuple starting at node $u$ as $p$, i.e., the number of 2-tuples starting from $u$ that has the same isomorphic type as $(u,v)$ in $s$. Then the number of $s$ that passes node $u$ is $\sum_{v \in N(u)}n_{uv} / k$. Take the counting of 3-cycles $s = \\{u, v, w\\}$ that passes $u$ as an example, $k$ is 2 since in $s$, $(u, v)$ and $(u, w)$ have the same isomorphic type as $(u,v)$ ($(u,w)$, resp.). Considering that there is a cycle that passes $(u, v)$ ($(u, w)$, resp.), the number of 3-cycles that passes $u$ is $(1+1)/2=1$.
>
> > Q4. $\dots$ Furthermore, I noticed there is an additional counting experiments on ZINC in the appendix, but why is it limited to cycle counting?
>
> **A6.** We report the number of these substructures in ZINC. Specifically, we report the average number of these substructures per graph (denoted as "graph") and the average number of these substructures that pass each node (denoted as "node"). The statistics are shown below.
>
> |ZINC |3-cycle | 4-cycle | 5-cycle | 6-cycle | tailed triangle | chordal cycle | 4-clique | 4-paths | triangle-rectangles |
> |  :----: |  :----: | :----: | :----: | :----: | :----: | :----: | :----: | :----: | :----: |
> |graph |  6.41e-2  | 1.60e-2 | 8.52e-1 | 1.80 | 8.79e-2 | 0 | 0 |  55.15 | 0 |
> | node | 8.30e-3 | 2.76e-3 | 1.84e-1 | 4.67e-1 | 3.80e-3 | 0 | 0 | 4.76 | 0 |
>
> We observe that ZINC may not be optimally suited for substructure counting tasks. A significant limitation is (1) the absence of certain substructures, such as 4-cliques, chordal cycles, and triangle-rectangles; (2) many substructures seldom exist in ZINC, such as the 3-cycles and 4-cycles. Therefore we put it in the appendix and do not report the performance on graphlets.
>
>
> > b. The time comparison on ZINC and OGB is presented in the main paper without reporting the results on those datasets in the main paper. Please move the results on these datasets in the main paper.
>
>
> **A7.** Thanks for your suggestion. It is temporarily not added due to space limits. We will add it to the main paper in the revised manuscript.
>
>
> > c. Why results on ZINC do not include the std or average across seeds? And why do you use a graph transformers? What are the results with a GNN as a backbone model?
>
>
> **A8.** Actually, the reported result is the mean MAE score across ten individual runs. We will add the std score in the revised manuscript. Specifically, ESC-GNN reaches an MAE score of 0.086$\pm$0.003 on ZINC with the graph transformer as the backbone network ((now an MAE score of 0.078$\pm$0.004, after adding an MLP layer after the initial embedding layer. A detailed analysis is available in “**Analysis on the real-world benchmarks**” in our response to Reviewer aWFU)). In addition, it reaches an MAE score of 0.096$\pm$0.006 with GIN as the backbone network. Note that the proposed encoding can significantly boost the performance for both backbone models.
>
> The reason for choosing the graph transformer as the backbone is based on empirical evidence. Previous works have shown that densely connected models such as the graph transformers (Ramp´aˇseketal.,2022.) or global information such as the laplacian eigenvectors (Lim et al. 2022) can boost the performance on ZINC. Since laplacian-based eigenvectors may violate permutation equivariance, we adopt the graph transformer without laplacian-based positional encoding as the backbone model.

---

> > ### Comment · Reviewer_p2Va · 2023-11-19
> >
> > I sincerely thank the authors for answering my questions, I still have one comment.
> >
> > **Regarding W1, permutation equivariance.** I understand the authors' idea and I agree that if that is the case permutation equivariance is not lost. However, I think this point is not explained in the paper, and Figure 1 might be misleading. Since $s_{uv}$ and $s_{vu}$ might be different, then shouldn't you show two directed edges for each pair of nodes that were connected in the original graph? For example, consider the undirected edge between $v_1$ and $v_2$ in Figure 1 (left). Then, I think the graph on the right should contain two directed edges, one from $v_1$ to $v_2$ augmented with $s_{v_2v_1}$ and the other from $v_2$ to $v_1$ augmented with $s_{v_1v_2}$.

---

> ### Author Response · Authors · 2023-11-20
>
> Thanks for your suggestions on clarifying the permutation equivariance of the proposed model. We totally agree that it is very important and will integrate your suggestions into our revised manuscript. An example figure is available in the revised manuscript.

---

### Official Review · Reviewer_qaTB · 2023-11-01

**Soundness:** 2 fair
**Presentation:** 2 fair
**Contribution:** 1 poor
**Rating:** 3
**Confidence:** 3

**Summary:**

The authors seek to understand the expressive power of GNNs vis-a-vis counting substructures. Such a study has already been done using subgraph enhanced GNNs. Since existing subgraph-enhanced GNNs are inherently not scalable (they look at all subgraphs of fixed size), the authors seek to circumvent this problem by devising pre-computed structural embeddings which avoid a brute-force aggregation over all subgraphs.

**Strengths:**

The authors inject hand-crafted structural information about subgraphs (degree encoding, node-level encoding, edge-level distance encoding) into a given graph, which allows a standard GNN on the graph to count small substructures such as 4-cycles, 4-clique and 3-paths. This allows them to avoid doing subgraph enhancement, which is usually expensive because one has to brute-force iterate over all subgraphs.

**Weaknesses:**

1. It is not true that all subgraph-enhanced GNNs suffer from scalability issues. Currently, there exist subgraph-enhanced GNNs which do not brute-force search over all subgraphs, instead trying to learn which subgraphs are relevant for enhancement: see Qian (2022).

Of course, sampling may not lead to "theoretically provable" counting power, but such theoretical results about counting power have limited relevance since the subgraphs being counted are really small and hence this is mainly a question of practical nature.

2. The results in Table 1 are not strong enough even if one considers the scalability gains due to hand-crafted embeddings.
The drop in performance as one goes to 5-cycles and 6-cycles is quite severe, indicating poor generalization.

I am not sure if the paper provides any substantial research with potential for impact, mainly because of the hand-crafted nature of the proposed models based on ad-hoc theoretical arguments which have little value in the general case and are useful only in extremely specific instances (subgraphs of size at most 3 to 4).

**Questions:**

1. Have the authors compared their results to more efficient subgraph-enhancement algorithms such as Qian (2022)?

2. (Section 4.) What are "globally expressive models"?

3. In Table 1, the column for "3-cycles" has all successful entries less that 0.001. ESC-GNN shows an error of 0.0074, yet it is in the same bracket. Can you explain how the cut-off of 0.01 for MAE was chosen?

4. "In conclusion, subgraph GNNs rooted at k-tuples with backbone GNN as powerful as m-WL can reach a similar counting power to (m + k)-WL while being much more efficient." What values of k and m do you use for experiments?

5. Section 4.2: "Subgraph GNNs have long been used to count substructures. Existing works mainly focus on counting certain types of substructures, e.g., walks (You et al., 2021) and cycles (Huang et al., 2023) and do not relate subgraph GNNs with substructure counting in a holistic perspective." Does an incremental extension to cliques/paths of size <=4 really make your framework holistic?

---

> ### Author Response · Authors · 2023-11-18
> **Response to Reviewer qaTB (1/2)**
>
> Thank you for your valuable feedback and insightful suggestions. Your comments have been diligently considered, and we will integrate your suggestions into our revised manuscript.
>
>
> > W1. It is not true that all subgraph-enhanced GNNs suffer from scalability issues. Currently, there exist subgraph-enhanced GNNs which do not brute-force search over all subgraphs, instead trying to learn which subgraphs are relevant for enhancement: see Qian (2022).
> Of course, sampling may not lead to "theoretically provable" counting power, but such theoretical results about counting power have limited relevance since the subgraphs being counted are really small and hence this is mainly a question of practical nature.
>
> > Q1. Have the authors compared their results to more efficient subgraph-enhancement algorithms such as Qian (2022)?
>
>
> **A1.** As you mentioned, the subgraph sampling strategy accelerates subgraph GNNs while losing the theoretical counting power/expressiveness. To assess the practical implications of this trade-off, we conduct an empirical evaluation of the performance and efficiency of OSAN on ZINC and OGBG-HIV, and report the results in the table below. In the table, “Preprocess” denotes the time (second) to preprocess the data (generate subgraphs), and “Run” denotes the total time (second) to train and test. For the configuration of the subgraph sampling strategy, we adopt the official config file from https://github.com/chendiqian/OSAN/blob/master/configs/ogbg-molhiv/node_select/sel25_subgraph3_imle.yaml (OGBG-HIV) and https://github.com/chendiqian/OSAN/blob/master/configs/zinc/imle_with_esan_model/khop3_subgraph_10.yaml (ZINC).
>
> We observe that despite employing a subgraph sampling strategy, OSAN is still not faster than ESC-GNN. This is because ESC-GNN only needs to run message passing on the whole graph, while OSAN needs to run message passing on a collection of subgraphs (even if the number of subgraphs is restricted). In addition, ESC-GNN and even MPNN outperform OSAN on both datasets, showing that the subgraph sampling strategy not only compromises theoretical counting power but also empirically diminishes the representational power of the model. The evidence from these comparisons underscores the superior efficiency and effectiveness of ESC-GNN in handling graph data.
>
>
>
> | ZINC | MAE| Preprocess| Run |
> |  :----: |  :----: | :----: | :----: |
> | MPNN |  0.163$\pm$0.004  | 6.2 | 1945.0 |
> | OSAN | 0.168$\pm$0.006|  182.8 | 8913.2 |
> | ESC-GNN | 0.086$\pm$0.003 | 362.4 | 2872.2 |
>
>
>
>
> | OGBG-HIV |AUCROC| Preprocess| Run |
> |  :----: |  :----: | :----: | :----: |
> | MPNN |  77.07$\pm$1.49  | 2.7 | 6296.8 |
> | OSAN | 75.96$\pm$1.34 | 6.6 | 7980.1 |
> | ESC-GNN | 78.62$\pm$1.06 | 1782.5 | 6301.0 |

---

> > ### Author Response · Authors · 2023-11-18
> > **Response to Reviewer qaTB (2/2)**
> >
> > > W2. The results in Table 1 are not strong enough even if one considers the scalability gains due to hand-crafted embeddings. The drop in performance as one goes to 5-cycles and 6-cycles is quite severe, indicating poor generalization.
> > I am not sure if the paper provides any substantial research with potential for impact, mainly because of the hand-crafted nature of the proposed models based on ad-hoc theoretical arguments which have little value in the general case and are useful only in extremely specific instances (subgraphs of size at most 3 to 4).
> >
> > > Q5. Section 4.2: "Subgraph GNNs have long been used to count substructures. Existing works mainly focus on counting certain types of substructures, e.g., walks (You et al., 2021) and cycles (Huang et al., 2023) and do not relate subgraph GNNs with substructure counting in a holistic perspective." Does an incremental extension to cliques/paths of size <=4 really make your framework holistic?
> >
> >
> > **A2.** First, our framework can count substructures with more than 4 nodes. For example, stars of arbitrary size and substructures like the tailed 4-cliques. While models such as PPGN (Maron et al.2019) can count up to 3-clique and up to 5-path, NGN (Zhang \& Li, 2021) can count up to 3-clique and up to 3-path, I$^2$-GNN (Huang et al. 2023) can count up to 4-clique and up to 5-path, they still make great contributions to the community.
> >
> > In addition, the proposed encoding is a general distance encoding, and not designed to explicitly count these substructures. It somewhat contains more information than the number of specific substructures. For example, we have proven that the number of 4-cycles can be computed using the distance information. Conversely, deducing distance information solely from the count of 4-cycles is impracticable, given the multiplicity of configurations that can form a 4-cycle.
> >
> >
> > Furthermore, in terms of empirical evidence. There are works, such as the GSN (Bouritsas et al) that explicitly enhance GNNs with the number of substructures. This includes some substructures that are beyond the counting power of our model and other subgraph GNNs.  However, when evaluated on real-world datasets, GSN achieves an MAE score of 0.115 on ZINC, whereas ESC-GNN attains an MAE score of 0.086 (now an MAE score of 0.078, after adding an MLP layer after the initial embedding layer. A detailed analysis is available in “**Analysis on the real-world benchmarks**” in our response to Reviewer aWFU). Similarly, utilizing the same backbone (GIN), GSN records an AUCROC score of 77.99 on OGBG-HIV, whereas ESC-GNN demonstrates a higher AUCROC score of 78.62 (now an AUCROC score of 79.21). This empirically shows the superiority of the proposed encoding.
> >
> >
> >
> > > Q2. (Section 4.) What are "globally expressive models"?
> >
> >
> > **A3.** It denotes high-order WLs, such as 3-WL/4-WL.
> >
> >
> > > Q3. In Table 1, the column for "3-cycles" has all successful entries less that 0.001. ESC-GNN shows an error of 0.0074, yet it is in the same bracket. Can you explain how the cut-off of 0.01 for MAE was chosen?
> >
> >
> > **A4.** First, we note that MPNNs and graph transformers (with positional encodings) cannot successfully count 3-cycles, therefore, “the column for "3-cycles" has all successful entries less that 0.001” is not true. In addition, the evaluation criterion follows (Huang et al. 2023). It is reasonable since we can directly use a rounding function to obtain the ground truth with such a low MAE score. Furthermore, we have proven in Section 5 that ESC-GNN can theoretically count 3-cycles.
> >
> >
> >
> > > Q4. "In conclusion, subgraph GNNs rooted at k-tuples with backbone GNN as powerful as m-WL can reach a similar counting power to (m + k)-WL while being much more efficient." What values of k and m do you use for experiments?
> >
> > **A5.** $k$ and $m$ are both set to 2.

---

> > > ### Comment · Reviewer_qaTB · 2023-11-19
> > >
> > > Thanks for your detailed answer.

---

> > > > ### Author Response · Authors · 2023-11-20
> > > >
> > > > Thank you for taking the time to read our response. We are happy to assist with any additional questions or address any concerns you may have. Please feel free to reach out to us.

---

### Official Review · Reviewer_7TFC · 2023-11-01

**Soundness:** 2 fair
**Presentation:** 3 good
**Contribution:** 2 fair
**Rating:** 5
**Confidence:** 3

**Summary:**

In the first part of the paper the authors generalize subgraph GNNs by allowing the use of higher-order GNNs on rooted subgraphs (so far, classical GNNs were considered on subgraphs). Corresponding theoretical results related to counting subgraphs are listed, these are easy generalizations of known results. In the second part of the paper, which is a bit orthogonal to the first part, a subgraph encoding technique is used to transform a graph into an edge weighted graph, on which a standard GNN is applied. It is shown that thanks to the preprocessing and encoding, more subgraphs can be counted than without this extra information.

**Strengths:**

1. The investigation of counting abilities of GNNs is important for understanding their expressive power.

2. The generalisation of subgraph GNNs to those that than can leverage higher-order GNNs is a sensible
extension from a theoretical point of view.

3. The idea to augment the input graph with information about subgraphs, followed by running a GNN is
a sensible data augmentation technique.

4. Theoretical results complement the proposed method.

**Weaknesses:**

1. The bulk of the paper advocates higher-order GNNs but then the proposed method is the application of a standard GNN on an augmented graph? There is a bit of a mismatch between theory and the proposed method.

2. The proposed method seems very related to approach by Bouritsas et al and Barceló et al in which subgraph information is used (isomorphism, homomorphism) alongside classical GNNs.

3. It is unclear what theoretical justifications of the proposed encoding method.

**Questions:**

**Q1** Please explain how Section 4 and Section 5 connect to each other.

**Q2** What is the rationale behind the structural encoding presented in section 5. What guarantees does it give? Or other encoding methods possible? (a la molecular finger printing).

**Q3** The proposed method uses handcrafted features (as part of encoding). How does it related to the work by Bouritsas et al in which edges carry counts of subgraphs?

---

> ### Author Response · Authors · 2023-11-18
> **Response to Reviewer 7TFC**
>
> Thank you for your valuable feedback and insightful suggestions. Your comments have been diligently considered, and we will integrate your suggestions into our revised manuscript.
>
>
>
> > W1. The bulk of the paper advocates higher-order GNNs but then the proposed method is the application of a standard GNN on an augmented graph? There is a bit of a mismatch between theory and the proposed method.
>
> > Q1 Please explain how Section 4 and Section 5 connect to each other.
>
>
> **A1.** In Section 4, we show that subgraph GNNs are nearly as powerful as $k$-WL in terms of counting substructures, and the key reason is the integration of distance information within the subgraph framework. However, a notable limitation of subgraph GNNs is their suboptimal efficiency. To address the limitation, in Section 5, we accelerate them by pre-computing the distance information and enhancing MPNNs with the information. We then theoretically (Section 5) and empirically (Section 6) evaluate the proposed model and observe that it preserves the representation power of subgraph GNNs. In other words, the proposed model can be viewed as an efficient subgraph GNN which successfully preserves the representation power of classic subgraph GNNs.
>
>
>
>
> > W2. The proposed method seems very related to approach by Bouritsas et al and Barceló et al in which subgraph information is used (isomorphism, homomorphism) alongside classical GNNs.
>
> >Q3 The proposed method uses handcrafted features (as part of encoding). How does it related to the work by Bouritsas et al in which edges carry counts of subgraphs?
>
>
> **A2.** It is important to note that (Bouritsas et al) and (Barceló et al) explicitly encode the number of substructures to enhance GNNs. The approach heavily relies on domain-specific knowledge and the desired substructures may significantly vary across different tasks. In contrast, we encode the more general distance information of subgraphs, which enables plain GNNs to count many substructures without the need to manually determine which substructures to include. In addition, the general distance information contains more structural information than the number of specific substructures. For instance, we have proven that the number of 4-cycles can be computed using the distance information. Conversely, deducing distance information solely from the count of 4-cycles is impracticable, given the multiplicity of configurations that can form a 4-cycle. The superiority is also justified by empirical evidence.
>
> Regarding the real-world dataset, (Bouritsas et al) achieves an MAE score of 0.115 on ZINC, whereas ESC-GNN attains an MAE score of 0.086 (now an MAE score of 0.078, after adding an MLP layer after the initial embedding layer. A detailed analysis is available in “**Analysis on the real-world benchmarks**” in our response to Reviewer aWFU). Similarly, utilizing the same backbone (GIN), (Bouritsas et al) records an AUCROC score of 77.99 on OGBG-HIV, whereas ESC-GNN demonstrates a higher AUCROC score of 78.62 (now an AUCROC score of 79.21). These results highlight the superiority of our proposed model.
>
>
>
> > W3. It is unclear what theoretical justifications of the proposed encoding method.
>
>
> > Q2 What is the rationale behind the structural encoding presented in section 5. What guarantees does it give? Or other encoding methods possible? (a la molecular finger printing).
>
>
> **A3.** The proposed encoding is directly motivated by the theoretical results in Section 4, where we prove that the distance information within subgraphs is the key to boosting the counting power of subgraph GNNs. Building upon this theoretical foundation, we design the proposed encoding, as detailed in Section 5, and prove that it preserves essential structural information. This includes the ability to count substructures and differentiate between non-isomorphic graphs, thereby preserving the representation power of subgraph GNNs. It accelerates subgraph GNNs since it does not need to run message passing among all subgraphs.
>
> To provide an intuitive example, consider the task of counting 3-cycles. Subgraph GNNs rooted on nodes can first compute the number of triangles that pass each node $u$ as the number of edges whose both endpoint nodes are with distance “1” to $u$. It then aggregates the numbers of all nodes and computes the number of 3-cycles within the graph. Notice that the distance information is actually contained in our proposed encoding.
>
>
> In terms of the fingerprint (e.g., the one that boosts the performance on OGBG-HIV), we note that this fingerprint encompasses numerous molecular structures, a collection of which are identifiable (or countable) by subgraph GNNs. While it is feasible to incorporate these molecular structures within our proposed encoding framework, we do not include the information in our initial experiments since such information is excessively domain-specific, which could potentially limit the general applicability of the proposed model.

---

> > ### Comment · Reviewer_7TFC · 2023-11-19
> > **Reply to the authors**
> >
> > Thanks you for the detailed responses. It would be great if you can incorporate any clarification give here (and also to the other reviewers) in the final version (main or supp material).

---

> ### Author Response · Authors · 2023-11-20
>
> Thanks for your suggestions and time one reading the response. We will integrate both your suggestions and the suggestions from other reviewers into our revised manuscript.

---

### Meta-Review · Area_Chair_HLyM · 2023-12-15

**Metareview:**

Three reviewers tended more or less strongly toward rejecting the paper. Even the more positive reviewer expressed and maintained during the rebuttal concerns about the strength of the approach and the significance of the theoretical analyses. Another reviewer was not sure if the paper provides any substantial research with potential for impact due to the hand-crafted nature of the proposed models based on ad-hoc theoretical arguments which have little value in the general case and are useful only in extremely specific instances. The paper also misses a comparison to highly related work such as Qian (2022). Overall, the reviewers agree that the paper is not ready for acceptance.

**Justification For Why Not Higher Score:**

Several significant weaknesses in the theoretical results and questions about the practical significance.

**Justification For Why Not Lower Score:**

-

---

### Decision · Program_Chairs · 2024-01-16

Reject